

# Monsoon Asia Rice Calendar: a gridded rice calendar in monsoon Asia based on Sentinel-1 and Sentinel-2 images

Xin Zhao[1], Kazuya Nishina[1], Haruka Izumisawa[2], Yuji Masutomi[3], Seima Osako[4], Shuhei Yamamoto[4]

[1]Biogeochemical Cycle Modeling and Analysis Section, Earth System Division, National Institute for Environmental Studies, 16-2 Onogawa, Tsukuba, Ibaraki, 305-8506, Japan
[2]Faculty of Life and Environmental Sciences, University of Tsukuba, 1-1-1 Tennodai, Tsukuba, Ibaraki, 305-8572, Japan
[3]Asia-Pacific Climate Change Adaptation Research Section, Center for Climate Change Adaption, National Institute for Environmental Studies, 16-2 Onogawa, Tsukuba, Ibaraki, 305-8506, Japan
[4]DATAFLUCT, Inc., 1-19-9 Dogenzaka, Shibuya, Tokyo, 150-0043, Japan

*Correspondence to*: Xin Zhao (zhao.xin@nies.go.jp) and Kazuya Nishina (nishina.kazuya@nies.go.jp)

**Abstract.** An accurate and spatially explicit large-scale rice calendar can enhance understanding of agricultural practices and their ecological services, particularly in monsoon Asia. However, currently available global- or continental-scale rice calendars suffer from coarse resolution, poor recording, and outdated information, which do not provide detailed and consistent information on rice phenology. To address this limitation, this study mapped a new (2019 to 2020) gridded (0.5° × 0.5° resolution) rice calendar for monsoon Asia based on Sentinel-1 and Sentinel-2 satellite images. The novelty of this rice calendar lies in the development of a consistent optimal methodological framework that allows spatially explicit characterization of the rice transplanting date, harvest date, and number of rice croppings. The methodological framework incorporates two steps: (1) detection of rice phenological dates and number of rice croppings through combination of a feature-based algorithm and the fitted Weibull function, and (2) spatio-temporal integration of the detected transplanting and harvest dates using von Mises maximum likelihood estimates. Results revealed that the proposed rice calendar can accurately identify the rice phenological dates for three croppings in monsoon Asia. When compared with single rice data from the census-based RiceAtlas rice calendar, the proposed rice calendar outperformed the MODIS-based RICA rice calendar. It exhibited bias of 4 and −6 days for transplanting and harvest dates, respectively, with marked improvement in MAE by 10 and 15 days, and in RMSE by 6 and 15 days for transplanting and harvest dates, respectively. In total, the proposed rice calendar can detect single, double, and triple rice cropping with area of $5.3 \times 10^6$, $4.5 \times 10^6$, and $0.9 \times 10^6$ km$^2$, respectively. This novel gridded rice calendar fills the gaps in finer-resolution rice calendars across major global rice production areas, facilitating research on rice phenology that is relevant to the climate change.

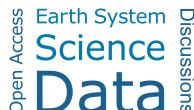

# 1 Introduction

A rice calendar records a series of phenological dates for rice growth and indicates the number of times that rice is grown in a year (Mishra et al., 2021; Zhao et al., 2023). Rice calendars provide critical information that contributes to agricultural management, crop production prediction, and estimation of greenhouse gas (GHG) emissions (Laborte et al., 2017; Portmann et al., 2010; Sacks et al., 2010). Specifically, concern regarding the negative impacts of rice cultivation is increasing because irrigated rice paddy field is an important source of anthropogenic GHG emissions, contributing 8 % and 11 % of global methane and nitrous oxide emissions, respectively (Saunois et al., 2020; Jiang et al., 2019). The inundated period from transplanting date to harvest date derived from a rice calendar largely determines the quantity of GHG emissions (Ito et al., 2021). To accurately estimate GHG emissions related to rice cultivation and to establish appropriate reduction measures, a detailed rice calendar that depicts rice phenology dynamics is urgently needed, especially for monsoon Asia, which accounts for 87 % of the area of harvested rice globally and for 90 % of global rice production (FAOSTAT, 2022).

Existing approaches to rice calendar mapping can be grouped into three categories: those based on census data, those based on models, and those based on remote sensing images. The limited number of global rice calendars (e.g., SAGE (Sacks et al., 2010), MIRCA2000 (Portmann et al., 2010), and RiceAtlas (Laborte et al., 2017)) that are currently available, which rely on compilation of statistical data at national and/or sub-national scales. Model-based rice calendars (Waha et al., 2012; Elliott et al., 2015; Mathison et al., 2017; Iizumi et al., 2019) provide large-scale spatially explicit rice phenology information that is mainly based on climate data, but they are difficult to validate using earth observation data over such scales (Mishra et al., 2021). In contrast, remote sensing approaches that can provide consistent detection of large-scale rice phenological change over time have been used for rice calendar mapping with varying spatial coverage (e.g., global (Kotsuki and Tanaka, 2015), Asia (Mishra et al., 2021; Zhang et al., 2022), South and Southeast Asia (More et al., 2016), China (Luo et al., 2020), and Japan (Sakamoto et al., 2005)).

However, many challenges in rice calendar mapping using remote sensing hinder the production of accurate rice calendar. First, use of a coarse–moderate-resolution satellite sensor (e.g., AVHRR with approximately 5 km resolution and MODIS with 500 m resolution) or any single sensor (optical or synthetic aperture radar (SAR)) diminishes the accuracy of rice calendar mapping. The RICA rice calendar, produced using MODIS images, has the problem that the area of a rice paddy field is typically smaller than the 500 m resolution of the sensor (Mishra et al., 2021). Second, the algorithm that extracts rice phenological dates largely determines the accuracy of rice calendar mapping; currently, the rule-based algorithm is predominantly used for such purposes (Zhao et al., 2023). Phenological date detection depends on the turning points or key nodes of a vegetation index or backscattering (Xin et al., 2020). The smoothing method and the parameters adopted change the time series pattern, which consequently produces different thresholds, inflections, curvatures, or second-order derivatives of the time series for each vegetation index or backscattering. Furthermore, the available algorithm is only used at a specific





administrative coverage, such as ChinaCropPhen1km for China (Luo et al., 2020), and EVI-related for Japan (Sakamoto et al., 2005), and lacks applicability to larger areas. Some other algorithms have been proposed to map rice phenological dynamics over large areas (e.g., both PhenoRice (Mishra et al., 2021) and LAI-related (Zhang et al., 2022) for Asia). However, the rice calendars produced are based on the administrative scale that ignores the fine heterogeneity in rice phenology within administrative units. Third, determination of the number of rice croppings is frequently biased. For example, some recent studies focused only on main rice cropping (Zhang et al., 2022) or determined rice cropping during specific time windows (Mishra et al., 2021), thereby excluding rice grown in other periods. Additionally, rice sometimes grows across years because double or triple rice croppings, making it difficult to determine the actual number of rice croppings. Although methods have been proposed for extracting the number of croppings through growing season peak detection (Kotsuki and Tanaka, 2015; Yan et al., 2019), much effort is required to reduce the uncertainty of bias in the peak caused by ratoons and/or noisy data (Liu et al., 2020).

The combination of optical and SAR sensors benefits from the availability of Copernicus Sentinel-1 and Sentinel-2 satellite images with high spatial (10 m) and temporal (6 days for Sentinel-1, 5 days for Sentinel-2) resolution. This sensor combination is ideal for monitoring crop phenology because it offers precise and timely information on crop phenological variations; consequently, it has been widely used in recent relevant research (d'Andrimont et al., 2020). A feature-based algorithm has been proposed for rice phenology detection at the large scale (Zhao et al., 2023). The superiority of this algorithm lies in the use of backscattering (VH) and vegetation indices (Enhanced Vegetation Index (EVI) and Normalized Yellow Index (NDYI)) derived from Sentinel-1 and Sentinel-2 images, which reflect features related to rice cultivation such as flooding, maximum leaf area, and most yellowness around transplanting, heading, and harvest date. Additionally, this algorithm has successfully tracked rice phenological dates of different cropping systems and at different spatial scales (Zhao et al., 2023). Thus, the recognition of rice phenological dates is based on the feature-based algorithm. Meanwhile, a fitted six-parametric Weibull function has successfully been adopted to depict the growth development of phytoplankton (Rolinski et al., 2007) and vegetation (Maciel-Nájera et al., 2020; Muñoz-Salazar et al., 2022). Because variation of greenness is a reasonable indicator of crop intensity, the ability of a fitted Weibull function to fit the beginning, peak, and end of the greenness cycle allows it to capture the number of rice croppings. Different from most widely used peak greenness detection methods, which depend on thresholds for detection, the fitted Weibull function omits the noisy peak, which means it can track the shape of the vegetation index time series. Moreover, the fitted Weibull function has been packaged in the R software, making detection of the number of rice croppings automatic. Therefore, a feature-based algorithm combined with a fitted Weibull function is suitable for extracting rice phenological dates and the number of rice croppings.

The objective of this study was to develop a new gridded rice calendar that highlights the following features: (a) consistent detection using remote-sensing methods, (b) high spatial resolution (0.5°), (c) large-scale coverage (monsoon Asia), and (d) ability to extract multiple rice croppings. To achieve this goal, Sentinel-1 and Sentinel-2 satellite images with high spatio-

temporal resolution, spanning 2019 to 2020, were integrated within a novel methodological framework. This framework consists of two main steps: (1) detection of rice phenological dates and the number of rice croppings using a combination of a feature-based algorithm and a fitted Weibull function, and (2) spatio-temporal integration of detected phenological dates using von Mises maximum likelihood estimates. The resulting rice calendar was then evaluated against existing rice calendars. The findings of this study provide valuable insight into the methodological framework and rice calendar products, benefiting both crop calendar algorithm developers and end users.

## 2 Materials and methods

### 2.1 Study area

Figure 1 shows the study area in monsoon Asia, which covers the region of 10° S to 53.5° N, 61° E to 153° E.

Monsoon Asia accounts for the largest rice harvested area and the greatest volume of rice production globally (Zhang et al., 2020). The rice paddy fields of monsoon Asia are mainly on the Indo-Gangetic Plain, the Yangtze Plain, in the Ayeyarwady Delta region, and the Mekong Basin (Zhang et al., 2020). India and China have the two largest rice harvested areas covering 44 and 30 million ha, respectively, followed by Bangladesh, Thailand, Vietnam, Myanmar, Philippines, Cambodia, Pakistan, and Nepal (Fig. S1) (FAOSTAT, 2022).



**Figure 1** Location of the study area and distribution of rice paddy fields in monsoon Asia. Rice paddy field distribution map **(a)** was
obtained from Zhang et al., (2020), which was produced using MODIS images. Green areas indicate rice paddy fields. Gridded rice paddy
field map **(b)** shows the percentage of rice paddy field in 0.5° grids. Green gradient indicates variation in the percentage coverage of rice
paddy fields.



## 2.2 Data

### 2.2.1 Rice paddy field distribution map and sampling method

The rice paddy field distribution map adopted in this study is from a 500 m resolution map produced using MODIS images (Zhang et al., 2020) (Fig. 1a). This map effectively displays the presence and distribution of rice paddy fields over monsoon Asia. The reliability of this map is substantiated by its strong correlation with existing rice paddy field maps across diverse areas, including China, North Korea, South Asia, and Southeast Asia, with $R^2$ values ranging from 0.72 to 0.95. Furthermore,

this map aligns well with the area information obtained from FAOSTAT statistical data for each country (Zhang et al., 2020). Additionally, this map has been used to develop a feature-based algorithm for rice phenology detection (Zhao et al., 2023).

This rice paddy field distribution map was aggregated to a gridded map with 0.5° resolution (Fig. 1b). Within each 0.5° grid,

20 rice paddy fields were randomly selected to derive the average rice phenology for that grid (Xiao et al., 2021; Zhao et al., 2023). This sampling method effectively minimizes errors caused by misclassification of rice paddy fields by excluding outliers that deviate from the averaged rice phenology (Zhao et al., 2023).

### 2.2.2 Satellite data

All available images from Sentinel-1 and Sentinel-2 from 1 January 2019 to 31 December 2020 were used for generating

backscattering or vegetation index time series via the Google Earth Engine (GEE) (http://earthengine.google.com/) platform and the Google CoLaboratory platform. To overcome inherent speckle noise and overlapping observations of Sentinel-1 images, $3 \times 3$ pixels moving window filter and incidence angle processing were performed (Inoue et al., 2020). Invalid observations of Sentinel-2 images caused by clouds and cirrus were removed using cloud filtering (> 50 %) and the cloud-score method (QA60 quality assessment band with 60 m resolution) (Inoue et al., 2020). The VH C-band Ground Range

Detected images in the Interferometric Wide Swath mode were acquired with 6 day temporal resolution. Based on the Sentinel-2 Multispectral Instrument Level-1C top of atmosphere reflectance images, the EVI and the NDYI, based on the blue (B2), green (B3), red (B4), and NIR (B8) spectral bands with 10 m spatial resolution and 5 day temporal resolution, were calculated as follows:

$$EVI = \frac{2.5 \times (NIR - Red)}{NIR + 6 \times Red - 7.5 \times Blue + 1}, \qquad (1)$$

$$NDYI = \frac{(Green - Blue)}{(Green + Blue)}, \qquad (2)$$

The Locally Estimated Scatterplot Smoothing (LOESS) method was further adopted to smooth the time series data. The span value was assigned as 0.075 and 0.2 to depict VH and the EVI/NDYI time series pattern.



### 2.2.3 Reference rice calendars

There are three widely accepted rice calendars currently available. The RiceAtlas rice calendar provides the start, peak, and end of the transplanting date and harvest date, and the number of rice croppings at national or sub-national scales globally. It is based on compilation of multiple data sources that include census data, databases, publications, and reports (Laborte et al., 2017). The RICA rice calendar was generated using MODIS images to map the rice transplanting date, harvest date, and number of rice croppings at administrative units in Asia (Mishra et al., 2021). The SAGE rice calendar records the gridded rice transplanting date and harvest date of 2000 at 5 min spatial resolution, but only records two rice croppings (Sacks et al., 2010). Therefore, the RiceAtlas, RICA, and SAGE rice calendars were used in this study to evaluate the number of rice croppings. The RiceAtlas rice calendar, with its phenological date range, was used to assess the performance regarding transplanting date and harvest date. The performance of the proposed rice calendar in determining the transplanting and harvest dates was assessed using the coefficient of determination ($R^2$), bias error (Bias), Mean Absolute Error (MAE), and Root Mean Square Error (RMSE) which were calculated as follows:

$$R^2 = 1 - \frac{(\sum_{i=1}^{n}(y_i - \bar{y})(s_i - \bar{s}))^2}{\sum_{i=1}^{n}(y_i - \bar{y})^2 - \sum_{i=1}^{n}(s_i - \bar{s})^2} \tag{3}$$

$$Bias = \frac{1}{n}\sum_{i=1}^{n}(y_i - s_i) \tag{4}$$

$$MAE = \frac{1}{n}\sum_{i=1}^{n}|y_i - s_i| \tag{5}$$

$$RMSE = \sqrt{\frac{1}{n}\sum_{i=1}^{n}(y_i - s_i)^2} \tag{6}$$

where $y_i$ and $\bar{y}$ are the phenological dates from the proposed rice calendar for sample grid ($i$) and the corresponding mean value, respectively, $s_i$ and $\bar{s}$ are the phenological dates from the reference rice calendar for sample grid ($i$) and the corresponding mean value, respectively, and $n$ represents the number of sampled phenological dates.

### 2.3 Methodology

The overall methodology for rice calendar mapping, which is summarized in Fig. 2, can be divided into two steps. The first step is extraction of transplanting and harvest dates and detection of the number of rice croppings, depicted in Fig. 2 Step 1-1 as the algorithm for phenological dates and number of rice croppings detection and in Fig. 2 Step 1-2 as the process of phenological dates and number of rice croppings detection. The transplanting and harvest dates obtained in the first step require temporal and spatial integration for the generation of the rice calendar (Fig. 2 Step 2). The following sections provide elaboration on the major procedures involved in each step.



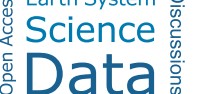

**Figure 2** Workflow for gridded rice calendar mapping based on satellite images. Step 1-1 depicts the algorithm, including transplanting and harvest dates extraction (Zhao et al., 2023), and the detection of number of rice croppings. Step 1-2 shows the process of extracting the number of rice croppings, transplanting date, and harvest date. Step 2 describes the generation of the rice calendar through temporal and spatial integration of the detected transplanting and harvest dates.



**2.3.1 Step 1: Algorithm and process for extraction of phenological dates and number of rice croppings**

**2.3.1.1 Algorithm (Step 1-1a) and process (Step 1-2b) for extraction of transplanting and harvest dates**

Flooding rice cultivation, as opposed to direct seeding, is common practice in Asia (Boschetti et al., 2017; Pandey et al., 2000) and accounts for more than 12 % of the global cropland area (FAOSTAT, 2020; Zhang et al., 2021a). This cultivation practice allows rice to grow in flooded soil, which presents a distinctive flooding signal that can be used for detection of rice transplanting date. Additionally, the time of rice harvest is characterized by irreversible yellowing of the leaves, resulting
from the rapid breakdown of chlorophyll and the photosynthetic apparatus (Zhang et al., 2021b) (Fig. 2 Step 1-1).

The feature-based algorithm was used on the smoothed VH, EVI, and NDYI time series data to capture the aforementioned phenological characteristics of rice crops (Zhao et al., 2023). The transplanting date was determined by identifying the minimum VH intensity from the shortest plants above the water surface. As the rice plants grow above the water surface and
interact with the incident radar signal, the VH intensity gradually increases (Torres et al., 2012). The harvest date was detected using the yellow signal derived from the NDYI, which employs a combination of green and blue bands to represent the balance between rice growth and senescence. Consequently, the NDYI value reaches a peak (approaches nearly 0 from negative values), indicating the maximum yellowness associated with the harvest date (Zhao et al., 2023) (Fig. 2 Step 1-1a).

The minimum VH and peak NDYI were detected within the time window (Fig. 2 Step 1-2b), indicating that only the minimum VH and the maximum NDYI values within the time window, before and after the EVI peak, can be identified as the transplanting date and harvest date, respectively. To identify the optimal time window for detection of the transplanting and harvest dates, the time window for detection of the minimum VH was set from 120 days before the date of peak EVI ($DOY_{EVI_{max}}$) to 45 days before the date of peak EVI, i.e., $\left[DOY_{EVI_{max}} - 120, \; DOY_{EVI_{max}} - 45\right]$ or from the first day of EVI
arc ($DOY_{EVI \, arc_{first \, day}}$) to 45 days before the date of peak EVI, i.e., $\left[DOY_{EVI \, arc_{first \, day}}, \; DOY_{EVI_{max}} - 45\right]$. The time window for detection of the peak NDYI was set from 13 days after the peak EVI date to 55 days after the date of peak EVI, i.e., $\left[DOY_{EVI_{max}} + 13, \; DOY_{EVI_{max}} + 55\right]$ or from 13 days after the peak EVI date to the last day of the EVI arc ($DOY_{EVI \, arc_{last \, day}}$), i.e., $\left[DOY_{EVI_{max}} + 13, \; DOY_{EVI \, arc_{last \, day}}\right]$. If the peak NDYI could not be obtained from those time windows, peak NDYI was identified using the peak EVI date ($DOY_{EVI_{max}}$) plus the corresponding difference days for each
rice cropping, as referenced in Zhao et al. (2013) (Fig. 2 Step 1-2b).

**2.3.1.2 Method (Step 1-1b) and process (Step 1-2a) for detecting the number of rice croppings**



The six-parametric Weibull function, $w(x) = \left(d + \exp\left(-\left(\frac{x}{e}\right)^f\right)\right) \times \left(1 - a \exp\left(-\left(\frac{x}{b}\right)^c\right)\right)$ (where $a$, $b$, $c$, $d$, $e$, and $f$ are the free parameters to be fitted) (Rolinski et al., 2007), can be used to identify the number of rice croppings by depicting an arc with the shape of downward-opening patterns from the smoothed EVI time series (Fig. 2 Step 1-1b). This fitted Weibull

function can be implemented using the peakwindow function in the "cardidates" package of R (Petzoldt et al., 2023; Rolinski et al., 2018; R Core Team, 2013) (https://cran.r-project.org/web/packages/cardidates/index.html) (Fig. S2a; Fig. 3). The rice cropping duration and its peak were determined as follows:

$$Cropping = peakwindow\ (x, y, mincut, minpeak) \tag{7}$$

where $y$ represents the variations of smoothed EVI time series values with respect to the date variable $x$.

To control the shape of the EVI arc, which represents the relative height between the neighbouring peaks and valleys, for the purpose of identifying rice cropping, the parameters *mincut* and *minpeak* were set to 0.9 (Figs. S2c and S3) and 0.6 (Figs. S2d and S3), respectively.

After application of the function (Eq. (7)), all available arcs of the smoothed EVI time series were then labelled, including

the start (start day of detected EVI arc, $DOY_{EVI\ arc_{first\ day}}$), peak (peak day of detected EVI arc, $DOY_{EVI_{max}}$), and end (end day of detected EVI arc, $DOY_{EVI\ arc_{last\ day}}$) of the arc, and the peak EVI value ($Value_{EVI_{max}}$) (Fig. 2 Step 1-2a).

This method can detect the EVI arc, even if it does not exhibit a complete downward-opening shape (Fig. S2b). This is because rice growth spans two years, and some days are not within the period of study, resulting in lack of EVI time series

data for those days. Based on the labelled EVI arc, all rice croppings were recognized, as the heading date through extraction of the $DOY_{EVI_{max}}$.



**Figure 3** Smoothed EVI time series and subsequent identification of the number of rice croppings at adjacent grids (32.25°N, 130.25°E, and 32.75°N, 130.25E°) across two years. Left column shows the smoothed EVI time series using the LOESS method. Black points and lines indicate the EVI value at specific dates and the smoothed EVI time series, respectively. Grey area indicates the 95 % confidence interval around the smoothed EVI time series. Right column displays the number of rice croppings detected using the fitted Weibull function implemented via the "cardidates" package in R. Blue, yellow, red, and black lines correspond to the detected first, second, third, and fourth arcs of the smoothed EVI time series.

## 2.3.2 Step 2: Temporal and spatial integration of detected transplanting and harvest dates for rice calendar generation

The transplanting and harvest dates across two years were detected using the algorithms and processes described above. However, the detected transplanting and harvest dates in each grid vary annually owing to different weather conditions, the effects of climate change, agricultural schedule adjustments, and the availability of satellite images. Additionally, the



detected transplanting and harvest dates for a certain cropping season in a grid can differ markedly from those of neighbouring grids, which might indicate detection errors. Therefore, temporal and spatial integration of the detected transplanting and harvest dates is a necessary step for generation of a multi-year averaged rice calendar.

To achieve this goal, the first step involved converting all the detected transplanting and harvest dates over two years into the
Day Of the Year (DOY) format, ranging from DOY 1 (1 January 2019) to DOY 730 (31 December 2020; 29 February 2020 was not considered for simplicity). Subsequently, the detected transplanting and harvest dates that occurred in 2020 (DOY 366 to DOY 730) were converted for consistency with the first year (DOY 1 to DOY 365) by subtracting 365 (Fig. 2 Step 2a). Finally, all the detected transplanting and harvest dates were converted to DOY values from 1 to 365 (Fig. 2 Step 2a).

The temporal and spatial integration of phenological dates should be conducted within specific periods of time. Typically, rice is cultivated up to three times annually in most areas (Mishra et al., 2021), which serves as a meaningful basis for dividing the year into three equal periods. Thus, the year was divided into three groups: Group 1: July–October (DOY 182 to DOY 304), Group 2: March–June (DOY 60 to DOY 181), and Group 3: November–February (DOY 305 to DOY 59) (Fig. 2 Step 2b). The detected phenological dates were assigned to the corresponding group based on the maximum number of days
from the transplanting date to the harvest date falling within that group (Fig. 2 Step 2c).

The phenological date DOY values represent circular data that exhibit periodicity or cyclicity (Mahan, 1991). The designation of high and low values is arbitrary (Berens, 2009). For example, DOY 365 is almost the same as DOY 1 with 1 day difference instead of a difference of 364 days. Adoption of statistical analysis commonly used with circular data can lead
to incorrect results, whereas the von Mises distribution $VM(\mu, \kappa)$ can display a circular unimodal distribution (Berens, 2009). The probability density function of the von Mises distribution can be expressed as follows:

$$p\,(x;\,\mu;\,\kappa) = \frac{1}{2\pi I_0(\kappa)}\,exp\,(\kappa\cos\,(x-\,\mu)) \tag{8}$$

where $I_0$ is the modified Bessel function of zero order, and for $-\pi \ll x \leq \pi, \kappa > 0$.

The availability of the "circular" R package (https://rdrr.io/rforge/circular/man/circular-package.html) is convenient for analysis of circular data. In the "circular" R package, the probability density function of the von Mises distribution can be displayed as follows:

$$qvonmises\,(x,\,mu,\,kappa) \tag{9}$$

where $mu$ is the mean direction of the distribution, and $kappa$ is a non-negative numeric value representing a concentration
parameter of the distribution; $mu$ and $kappa$ are correspond to $\mu$ and $\kappa$ in Eq. (8), respectively.



The circular data $x$ in Eqs. (8) and (9) denote the phenological date shown in DOY format. The DOY was converted to an angle value (degrees) (Fig. 2 Step 2d) (Franch et al., 2022) as follows:

$$DOY_{deg} = \frac{DOY}{365} \times 360 \tag{10}$$

where $DOY_{deg}$ represents the angular value of the DOY, and 365 denotes the number of equal interval date units representing one rotation around the circle.

The angle value of the DOY ($DOY_{deg}$) was then converted to the radian value of the DOY ($DOY_{rad}$) with interval $[-\pi, \pi]$ (Fig. 2 Step 2d) as follows:

$$DOY_{rad} = DOY_{deg} \times \frac{\pi}{180} - \pi = \frac{(DOY - 182.5) \times \pi}{182.5} \tag{11}$$

Then, $DOY_{rad}$ was input into the mle.vonmises function in the "circular" R package to obtain the parameters of the von Mises distribution via maximum likelihood estimates. For each group (i.e., Group 1, Group 2, and Group 3), the $DOY_{rad}$ of each grid and the eight neighbouring grids ($3 \times 3$ pixel window) across the two years were included as input for the mle.vonmises function (Fig. 2 Step 2d). Overall, 18 $DOY_{rad}$ values were used as follows:

$$res = mle.vonmises\ (DOY_{rad1}, DOY_{rad2}, DOY_{rad3}, \dots, DOY_{rad18}) \tag{12}$$

The parameters $DOY_{integrated}$ and $Var$ were derived from the mle.vonmises function (Eq. (12)), representing the value and variance of the phenological dates ($DOY_{rad}$), respectively, after performing temporal and spatial integration for each grid within each group:

$$DOY_{integrated} = res\$mu \tag{13}$$

$$Var = res\$kappa \tag{14}$$

However, this value and variance of the phenological date ($DOY_{integrated}$ and $Var$) is a radian value, which must be converted back to the DOY ($DOY_{mu}$ and $DOY_{var}$) as follows:

$$DOY_{mu} = DOY_{integrated} \times \frac{182.5}{\pi} + 182.5 \tag{15}$$

$$DOY_{var} = \frac{1}{Var} \times \frac{180}{\pi} / \frac{360}{365} = \frac{1}{Var} \times \frac{182.5}{\pi} \tag{16}$$

The integrated transplanting dates from all the groups (e.g., $DOY_{mu\_G1}$, $DOY_{mu\_G2}$, and $DOY_{mu\_G3}$) were then reordered according to their chronological order for each grid. Finally, cropping-based phenological dates were obtained through the conversion of the group-based phenological dates (Fig. 2 Step 2e).



## 3 Results and discussion

### 3.1 Transplanting date and harvest date

Based on the above methodological framework, rice calendars with two types of transplanting and harvest dates were obtained: a group-based calendar (Fig. 4) and a cropping-based calendar (Fig. 6). The group-based rice calendar was initially produced, displaying explicit transplanting and harvest dates for three groups (Fig. 4). The median transplanting dates across monsoon Asia for the three rice groups are DOY 182, 76, and 325, with standard deviations of 23, 24, and 60, respectively (Fig. 4). Similarly, the median harvest dates for the three groups are 281, 172, and 67, with standard deviations of 23, 27, and

56, respectively (Fig. 4). Because the three groups divide the year equally, the transplanting date and the harvest date both exhibit a mono-peaked distribution (Fig. S4). Moreover, the variance of the transplanting or harvest dates observed in each grid originates from analysis of 18 detected transplanting or harvest dates from its eight neighbours across the two years, thereby highlighting both its temporal and spatial variations. The variance in transplanting and harvest dates across monsoon Asia for the three groups is shown in Fig. 5. The variance is 8, 11, and 12 for the transplanting dates and 7, 11, and 15 for the

harvest dates (Fig. 5). These variances arise from interannual variation and spatial smoothing effect, and their small values indicate stability in phenological date extraction.





**Figure 4** Transplanting date and harvest date for the three groups. Upper, middle, and lower panels of the left column show the transplanting date for Group 1, Group 2, and Group 3, respectively. Upper, middle, and lower panels of the right column show the harvest date for Group 1, Group 2, and Group 3, respectively. Colour gradient from blue to red in the legend denotes the respective transplanting and harvest dates.



**Figure 5** Variance in transplanting date and harvest date for the three groups. Upper, middle, and lower panels of the left column show the variance in transplanting date for Group 1, Group 2, and Group 3, respectively. Upper, middle, and lower panels of the right column show the variance in harvest date for Group 1, Group 2, and Group 3, respectively. Colour gradient from blue to red in the legend denotes the respective variance in transplanting and harvest dates.






Then, the group-based transplanting and harvest dates were converted to the cropping-based format by reordering the detected transplanting dates for each grid. The cropping-based transplanting and harvest dates are in a common format that facilitates comparison with those of other rice calendars. The median transplanting dates across monsoon Asia for three rice croppings are DOY 154, 208, and 327, with standard deviations of 61, 68, and 27, respectively (Fig. 6). Similarly, the median harvest dates for three croppings are 253, 273, and 62, with standard deviations of 63, 111, and 47, respectively (Fig. 6). Owing to the large spatial coverage, the transplanting and harvest dates vary across different croppings, exhibiting a dual-peaked distribution (Fig. 8). The variance in transplanting dates for three croppings across monsoon Asia is 9, 10, and 12, while the variance in harvest dates is 8, 11, and 16 (Fig. 7).





**Figure 6** Transplanting date and harvest date for three rice croppings. Upper, middle, and lower panels of the left column show the transplanting date for Cropping 1, Cropping 2, and Cropping 3, respectively. Upper, middle, and lower panels of the right column show the harvest date for Cropping 1, Cropping 2, and Cropping 3, respectively. Colour gradient from blue to red in the legend denotes the respective transplanting and harvest dates.



**Figure 7** Variance in transplanting date and harvest date for three rice croppings. Upper, middle, and lower panels of the left column show the variance in transplanting date for Cropping 1, Cropping 2, and Cropping 3, respectively. Upper, middle, and lower panels of the right column show the variance in harvest date for Cropping 1, Cropping 2, and Cropping 3, respectively. Colour gradient from blue to red in the legend denotes the respective variance in transplanting and harvest dates.


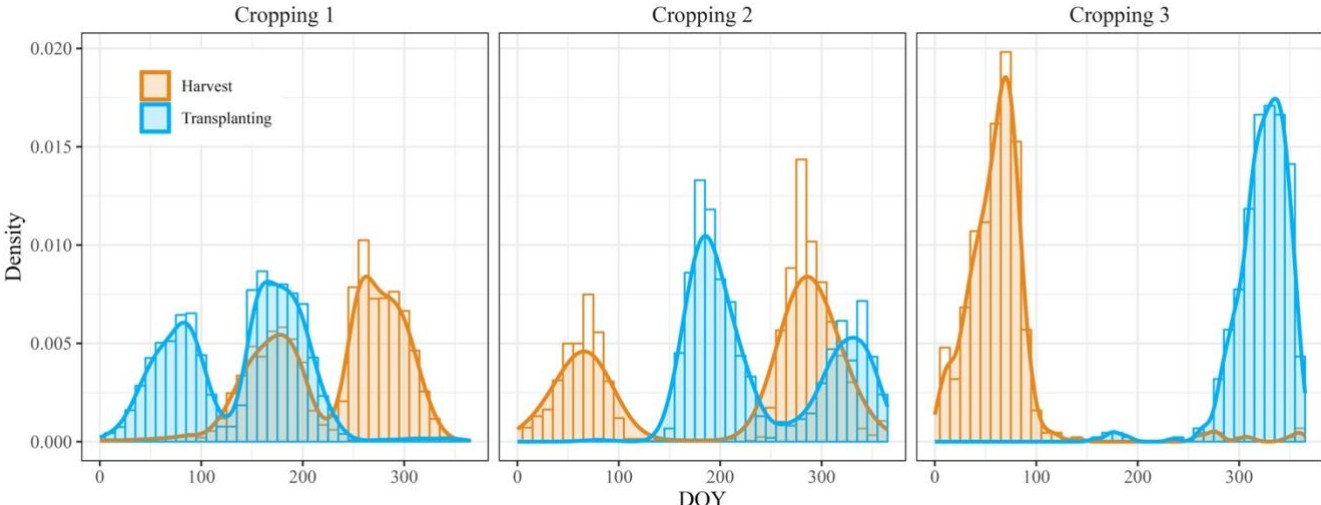

**Figure 8** Distribution of transplanting and harvest dates for three rice croppings. Blue and orange represent the transplanting date and the
harvest date, respectively.

There are difficulties in directly comparing transplanting and harvest dates with those from other rice calendars owing to differences in spatial resolution (grid versus administrative) and the identification of rice cropping sequences (Clauss et al., 2018). Thus, the RiceAtlas rice calendar, which has been rasterized to the same spatial resolution as that of the proposed rice
calendar (0.5°), was used to evaluate the performance in terms of single rice cropping for the transplanting and harvest dates. The transplanting dates of the proposed rice calendar are consistent with those of the RiceAtlas rice calendar, with Bias of 3.93 days, MAE of 16.38 days, and RMSE of 27.62 days (Fig. 9). Additionally, the harvest dates of the proposed rice calendar are correlated with those of the RiceAtlas rice calendar, with Bias of −5.76 days, MAE of 17.87 days, and RMSE of 28.32 days (Fig. 9). However, the presence of the same transplanting or harvest dates across large spatial areas in the
RiceAtlas rice calendar (Fig. S5) reduces its accuracy. Similarly, the RiceAtlas rice calendar has been used to evaluate the performance of the MODIS-based RICA rice calendar (Mishra et al., 2021). The proposed rice calendar outperforms the RICA rice calendar in terms of accuracy, with smaller MAE (26.41 days in the RICA rice calendar) and RMSE (34.20 days in the RICA rice calendar) in relation to transplanting dates, and almost half the MAE (33.20 days in the RICA rice calendar) and smaller RMSE (42.72 days in the RICA rice calendar) in relation to harvest dates (Mishra et al., 2021).


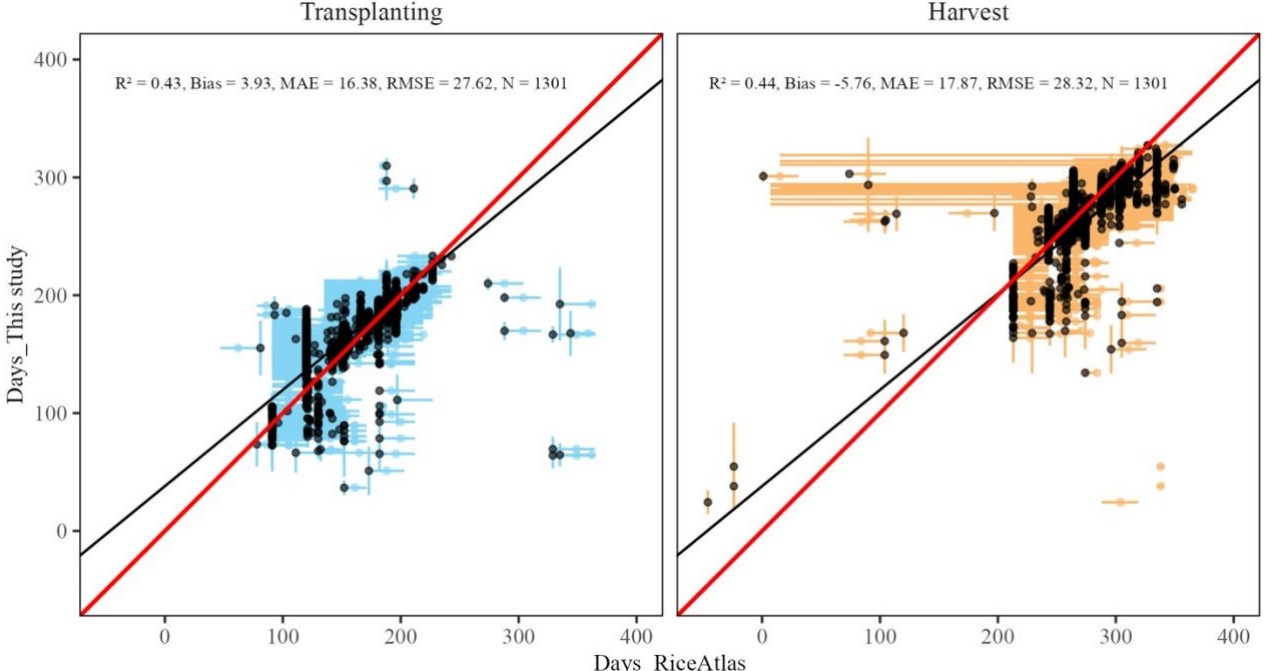

**Figure 9** Comparison of transplanting date and harvest date for single rice cropping between the proposed rice calendar and the RiceAtlas rice calendar. Blue and orange represent the transplanting date and harvest date, respectively; vertical lines denote the range of the transplanting and harvest dates of the proposed rice calendar; horizontal lines denote the range of the transplanting and harvest dates of the RiceAtlas rice calendar; dots denote the peak of the transplanting or harvest dates. Black dots denote the detected phenological day that falls within the transplanting or harvest ranges from the RiceAtlas rice calendar. Grey dotted line and red solid line represent the 1:1 line and regression, respectively.

## 3.2 Number of rice croppings

The number of rice croppings in the proposed rice calendar was obtained by counting the phenological dates for the three croppings (Fig. 10a). In comparison with the RiceAtlas (Fig. 10b), RICA (Fig. 10c), and SAGE (Fig. 10d) rice calendars based on the administrative scale, the proposed rice calendar shows the number of rice croppings per grid, which cannot be paralleled by the other rice calendars. Among the total of 4811 detected grids, 2728, 1644, and 439 grids were identified as single, double, and triple rice croppings, respectively. To compare the number of each rice cropping across the rice calendars

with different spatial resolutions, the area for each number of croppings was calculated (Fig. 11). In the area calculation, the variation of the area of each grid cell on the ellipsoidal earth (Fig. S8) was considered, as was the percentage coverage of rice paddy fields in each grid (Fig. 1b).

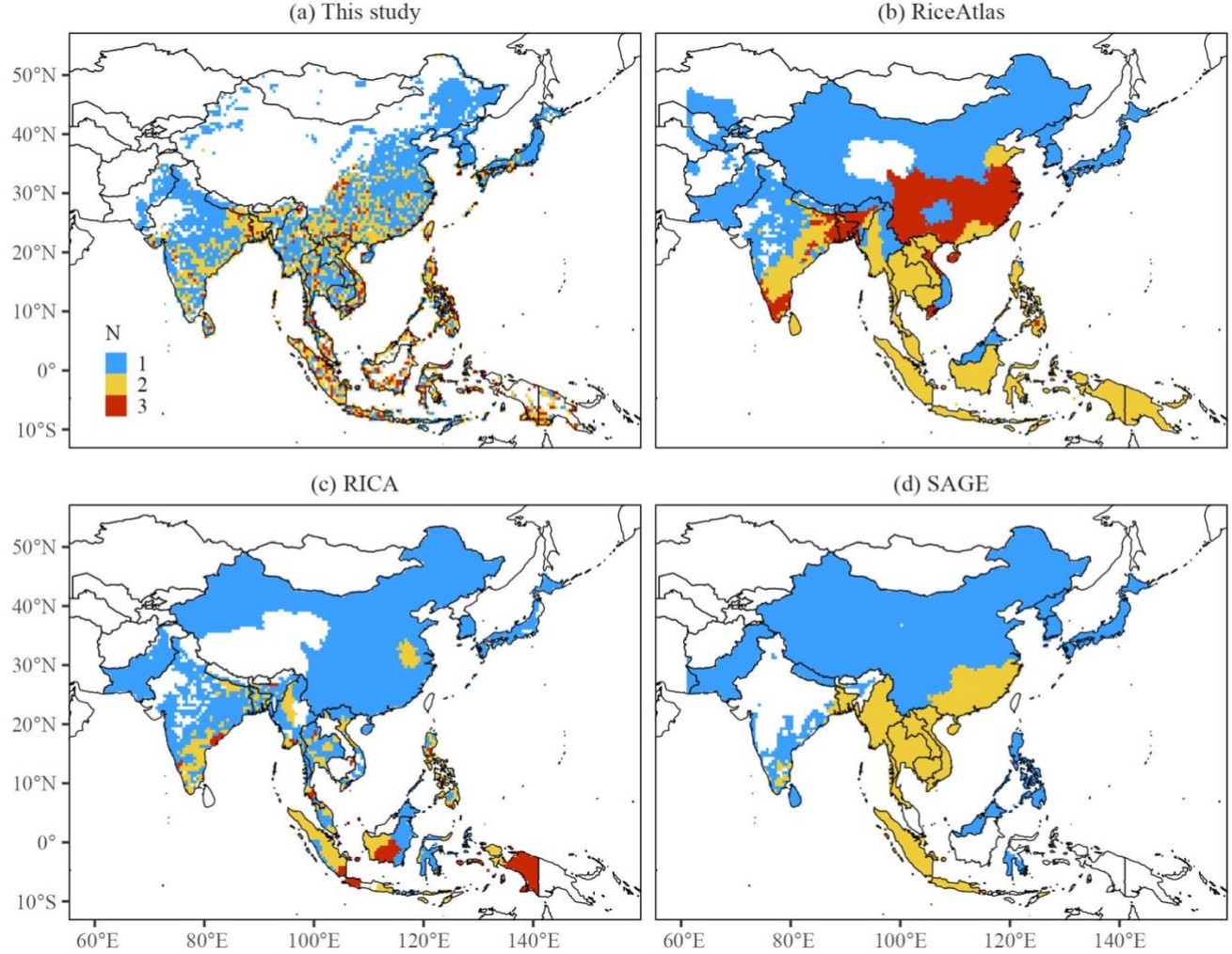

**Figure 10** Detected number of rice croppings of **(a)** the proposed rice calendar, **(b)** the RiceAtlas rice calendar, **(c)** the RICA rice calendar, and **(d)** the SAGE rice calendar. Blue, yellow, and red colours represent single, double, and triple rice cropping, respectively.

The areas covered by single, double, and triple rice croppings in the proposed rice calendar are $5.3 \times 10^6$, $4.5 \times 10^6$, $0.9 \times 10^6$ km², respectively (Fig. 11). The area covered by single rice cropping falls within the range of $2.4 \times 10^6$ (RiceAtlas rice calendar) to $6.5 \times 10^6$ km² (RICA rice calendar) (Fig. 11). The single rice cropping detection of the proposed rice calendar shows reasonable performance, with consistent detection across the north of the middle–lower reaches of the Yangtze River, North Korea, South Korea, and most of Japan when compared with the other three rice calendars (Fig. 10). The regions of Haryana, Himachal Pradesh, and Punjab in India, central, midwestern, and western Nepal, and Balochistan, the Federally Administered Tribal Areas, and the North-West Frontier Province in Pakistan were also identified as single rice cropping areas (Fig. 10). In fact, these regions were initially identified as having double rice cropping, but they are dominated by the

rice–wheat cropping system (Abrol, 1997; Dhanda et al., 2022; Ahmad and Iram, 2023). Therefore, in this study, wheat

cropping was removed within this region (Figs. 4–7, 10, 11, S9a, and S9b) (Abrol, 1997; Dhanda et al., 2022).

**Figure 11** Area of rice cropping of **(a)** the proposed rice calendar, **(b)** the RiceAtlas rice calendar, **(c)** the RICA rice calendar, and **(d)** the SAGE rice calendar. Blue, yellow, and red colours represent single, double, and triple rice cropping, respectively. Area calculation was
based on the percentage of rice paddy field map (Fig. 1b) and area of each grid cell on the ellipsoidal earth (Fig. S8).

The region of detected double rice cropping occupies a larger area in the proposed rice calendar than in the other three rice

calendars (Fig. 11a). Additionally, the area covered by triple rice cropping falls within the range of $0.5 \times 10^6$ (RICA rice

calendar) to $4 \times 10^6$ km$^2$ (RiceAtlas rice calendar). The proposed rice calendar successfully detects the mix of double and

triple rice croppings in Southeast Asia, including Vietnam, Malaysia, Indonesia, and the Philippines. The results align with

real-world observations. Double or triple rice croppings are mostly cultivated in Vietnam (Nguyen et al., 2012; Diem et al.,

2021). Two main croppings are cultivated in Malaysia (Fatchurrachman et al., 2022) and the Philippines (Laborte et al., 2012). Even on Java Island (Indonesia), the cultivation ranges from single to triple rice cropping systems (Ramadhani et al., 2020). In contrast, fewer areas of double rice cropping were detected in Myanmar, Thailand, Laos, and Cambodia using
remote sensing-based methods such as the proposed calendar (Fig. 10a) and the RICA rice calendar (Fig. 10c), in comparison with census-based rice calendars such as the RiceAtlas (Fig. 10b) and SAGE (Fig. 10d) rice calendars. The presence of a large amount of small but highly heterogeneous rice paddy fields limits accurate detection of the number of rice croppings using remote sensing (Mishra et al., 2021). Moreover, census-based rice calendars record the potential number of rice cropping, which means that some rice croppings might overlap or that triple rice cropping is determined even though
it accounts for only a small proportion of the rice cultivation activity within an administrative unit. This is a condition that does not occur in association with remote sensing-based rice calendars.

The proposed rice calendar extracts 9 % of triple rice croppings (Fig. 11a), which are scattered and distributed in South China, Southeast Asia, and India (Fig. 10a). This proportion is close to that of the RICA rice calendar (6 % in Fig. 11c), but
markedly lower than that of the RiceAtlas rice calendar (41 % in Fig. 11c). However, the larger percentage of triple rice croppings in the RiceAtlas rice calendar might be overestimated, especially in Northeast India and Bangladesh in areas of the lower Gangetic Plain. The double rice cropping system is predominant on the lower Gangetic Plain (Wang et al., 2020). In this area, rice cultivation occurs in one to three seasons, namely *Aus* (Mar/April/May to June/July), *Aman* (June/July/August to November/December), and *Boro* (November/December to January–April/May). Among the three seasons, *Aus* and *Aman*
are the dominant croppings (Gunna et al., 2014; Singha et al., 2019). Similarly, in South China, which is dominated by double rice cropping, early rice is transplanted at the end of April and harvested at the end of July, while late rice is cultivated from June to October (Chen et al., 2020).

### 3.3 Advantages of the proposed rice calendar

The aforementioned robustly supports the efficacy of the proposed rice calendar in depicting the detailed spatial variation of
rice phenology across a large area. Thus, it can be considered a reliable gridded rice calendar for monsoon Asia. The use of remote sensing-based methods provides precise and timely monitoring of the phenological condition and development of rice crops. Furthermore, the combination of Sentinel-1 and Sentinel-2 imagery contributes to the spatial explicitness of the rice calendar because the Sentinel satellites are considered to open a new era of dense and detailed observations that could overcome the long temporal frequency of Landsat images and the spatial resolution limitations of MODIS in mapping rice
calendars. Moreover, the Sentinel satellites were launched in 2014 and are scheduled to operate until 2030, thereby ensuring long-term continuous observation of rice phenology (Veloso et al., 2017).

A rice feature-based phenology algorithm (Zhao et al., 2023) was applied to Sentinel-1 and Sentinel-2 images to detect gridded rice transplanting and harvest dates (Fig. 2 Step 1-1a). This algorithm successfully extracted rice transplanting and



harvest dates from site to 0.5 grid-cell scale across monsoon Asia (Zhao et al., 2023). The robustness of validation at multiple spatial scales (Zhao et al., 2023), combined with reasonable performance in comparison with that of other rice calendars (Fig. 9), make the efficacy of the proposed rice calendar even more convincing. Furthermore, the proposed rice calendar outperforms the other rice calendars in terms of its algorithm for phenological date extraction. It overcomes the problem of overlap between rice croppings associated with census-based methods (such as RiceAtlas), and does not rely an

algorithm with constant threshold values set for large areas, as is the case with other remote sensing-based methods (such as RICA). In contrast, the feature-based algorithm is not limited by rice variety, management, and environmental factors. Instead, the features of flooding around the transplanting date and the most yellowness when harvested are extracted from the minimum VH and peak NDYI values, respectively, without threshold parameter setting for characterizing rice phenological variations.


Detection of the number of rice croppings is another import part of the methodological framework of the proposed rice calendar. The fitted Weibull function, implemented in the R package (Fig. 2 Step 1-1b), automatically detects the number of rice croppings based on the shape of the smoothed EVI time series, facilitating rapid and efficient rice calendar mapping. The shape-based detection avoids identification of the number of rice croppings based on peak detection or on the

occurrence of some certain phenological date within the rice season (e.g., flowering date, as in Mishra et al. (2021)). Additionally, the EVI shape-based detection allows identification of incomplete EVI arcs caused by non-continuous observations as one of the rice croppings (Fig. S2b).

Temporal and spatial integration of detected transplanting and harvest dates pose a great challenge owing to flexible

agricultural schedules and the availability of satellite imagery. This limitation restricts widespread application of remote sensing-based rice calendars. In this study, a new algorithm (Fig. 2 Step 2) was proposed to address this problem, which has long been a challenge in the preparation of previous rice calendars (Mishra et al., 2021). The use of von Mises maximum likelihood estimates produces the average of the transplanting and harvest dates for 18 grids ($3 \times 3$ grids $\times$ 2 years), taking the circular nature of phenological dates into special consideration (Fig. 2 Step 2d). This algorithm is of great benefit for

application in tropical areas where rice growth continues throughout the year, let alone temperate areas where rice growth occurs once a year. Additionally, the superiority of this algorithm lies in its ability to consider all rice croppings instead of excluding one of the rice cropping seasons through direct averaging based on administrative units. Furthermore, this algorithm improves the accuracy of the rice calendar by employing spatio-temporal integration, which reduces the presence of abnormal phenological dates.


The advantage of the above-mentioned algorithms (Fig. 2 Step 1, Step 2) contributes to the mapping of the gridded rice calendar. The proposed rice calendar fills the gaps in finer-scale rice calendars with continental coverage using remote sensing methods. It presents a highly patchy map of rice phenological information (Figs. 6 and 10a). The 0.5° resolution of





the proposed rice calendar is finer than that of other rice calendars, such as the RiceAtlas, RICA at sub-national scale, and
SAGE derived from sub-national data. This greatly reduces the bias error caused by assigning averaged rice phenology to an
administrative unit because there can be considerable differences in rice phenology within large administrative units (Franch
et al., 2022). Additionally, the proposed rice calendar displays the detailed distribution of rice paddy fields (Figs. 6 and 10a,
while previous rice calendars cover entire administrative areas, regardless of the small proportion of rice cultivation (Figs.
S5–S7 and 10b–d.

### 3.4 Uncertainty

Although the potential and advantages of the proposed gridded rice calendar for monsoon Asia have been highlighted, some
uncertainties remain.

(1)      *Limited experimental periods*. The proposed gridded rice calendar was produced based on detection during the
period 2019–2020. Although implementation of rice calendar mapping was facilitated by the GEE and Google CoLaboratory
platforms, generating detailed detection for two years ($127 \times 184 = 23{,}368$ grids $\times$ 2 years) still requires large computational
power. Increasing effort should be made in this regard, allowing for the use of a long-term time series of satellite images to
detect rice phenological dates. This, consequently, could result in a more stable and representative rice calendar, while
reducing the effects of climate change and/or agricultural schedule adjustments on rice calendar mapping.

(2)      *Errors produced through the spatial and temporal integration of detected transplanting and harvest dates*. The
proposed algorithm implemented spatial and temporal integration of detected transplanting and harvest dates (Fig. 2 Step 2),
which has often been underappreciated in the process of mapping rice calendars. However, one of the processes, i.e.,
grouping (Fig. 2 Step 2c), poses the risk of assigning single rice cropping seasons from two years into different groups.
Despite great effort having been made to overcome this problem, the possibility of overestimating the number of rice
croppings remains. The $Var$ parameter (indicating variance) derived from the mle.vonmises function (Eqs. (12) and (14)) is
seriously biased and requires bias-corrected estimates when the sample size is less than 16 (Best and Fisher, 1981). For this
reason, a $3 \times 3$ pixel window was used over the course of two years to produce 18 values. The selection of window size can
produce different phenological date values. Therefore, there is need to balance both the window size and the sample size in
the spatio-temporal integration of detected phenological dates.

(3)      *Accuracy of reference rice calendars*. Although the proposed rice calendar showed reasonable performance in
comparison with that of the other rice calendars, it should consider the accuracy of the reference rice calendars. In particular,
when evaluating transplanting and harvest dates against the RiceAtlas rice calendar, it is worth noting that the phenological
dates were sourced from census data, databases, publications, and report compilations (Laborte et al., 2017). Another
concern regarding the RiceAtlas rice calendar is the overlap of phenological dates between rice cropping seasons.
Additionally, the RiceAtlas, RICA, and SAGE rice calendars are based on the administrative scale, resulting in large spatial
coverage with only one recorded phenological date and number of rice croppings (Figs. S5–S7). It should also be mentioned
that some rice calendars, e.g., SAGE, are poorly documented and do not record triple croppings (Sacks et al., 2010).

(4)     *Overestimation of number of rice croppings caused by complex pattern of multiple-crop cropping systems.* Complex patterns of multiple-crop cropping systems refer to the cultivation of rice and other crops on the same land within a year, e.g., the middle rice cropping system (rice with wheat, barley, or rapeseed cropping systems) in East and Central China
(Chen et al., 2020), and the rice–wheat cropping systems on the Indo-Gangetic Plain (Abrol, 1997; Dhanda et al., 2022). In such systems, the growth of the other crop exhibits a similar pattern of a mono-peaked EVI time series and flood irrigation before sowing, as is the case with rice (Ahmad and Iram, 2023). This similarity of the signal often leads to misinterpretation of the other crop as another rice cropping. Although wheat cropping was manually excluded in this study in its primary cultivation areas (e.g., middle–lower reaches of the Indo-Gangetic Plain), some other areas might be affected by this
problem. Additionally, ratoon rice, excluded in this study by the parameter setting of the fitted Weibull function, might still be recognized in some circumstances as another rice cropping, making its total exclusion difficult (Fig. 3).

These uncertainties do not obscure the fact that this is a novel gridded rice calendar that provides more detailed rice phenology information, and could be input into ecosystem models for GHG emission evaluation and production prediction.
With the continued efforts of the research community to increase the spatio-temporal resolution of earth observational data, integrated use of the new rice paddy field distribution map, and implementation of new tools for improved analysis of huge satellite images, it should become feasible to produce more precise rice calendars at finer scale. Meanwhile, the methodological framework developed in this study for mapping the proposed rice calendar provides robust reference for mapping other crop calendars.

**4 Data availability**

The developed rice calendar described in the manuscript will be available at the Global Environmental Database (GED) once the curation process is complete, but is temporarily available at https://db.cger.nies.go.jp/MD/10.17595/2023XXXX.001.html.en (Zhao and Nishina, 2023) during the review process of this manuscript.

**5 Conclusions**

Given the absence of an updated global/continental-scale rice calendar that can explicitly depict spatial gridded transplanting date and harvest date information, and the number of rice croppings, this study developed a new gridded rice calendar for monsoon Asia with fine detail of rice phenology using a new methodological framework based on Sentinel-1 and Sentinel-2 images. Combination of a feature-based algorithm and a fitted Weibull function facilitates extraction of the transplanting and
harvest dates and detection of number of rice croppings, respectively. Subsequently, the detected transplanting and harvest dates were subjected to temporal and spatial integration to produce the rice calendar. The proposed rice calendar was found

sufficiently robust to map rice phenology more finely than that presented in other commonly used rice calendars, showing small Bias and improvement in both MAE and RMSE in terms of detection of transplanting and harvest dates. The proposed rice calendar could be used for global research on climate change and crop security, and the methodological framework could serve as a basis for producing large-scale mapping calendars for other crops.

**Author contributions**

KN conceived the conceptualization, acquired the funding, and supervised the project. XZ and KN developed the algorithm and curated the data. XZ, KN, and HI analysed the data. HI, SO, and SY contributed with satellite data analysis, and algorithm implementation. XZ and KN wrote the original draft, and XZ, KN, HI, YM, SO, and SY all reviewed the that draft.

**Competing interests**

The authors declare that they have no conflict of interest.

**Financial support**

This study was supported by the New Energy and Industrial Technology Development Organization (NEDO) (grant no. JPNP18016). This study was also partly supported by the Environment Research and Technology Development Fund (JPMEERF20202005) of the Environmental Restoration and Conservation Agency (ERCA).

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
