# Peer review of "Monsoon Asia Rice Calendar (MARC): a gridded rice calendar in monsoon Asia based on Sentinel-1 and Sentinel-2 images"

_Earth System Science Data, 2023_

## Author Comment (AC1)

**Response to the anonymous reviewer's comments**

*Anonymous referee #1:*

*Zhao et al. mapped a new (2019 to 2020) gridded (0.5° × 0.5° resolution) rice calendar for monsoon Asia based on Sentinel-1 and Sentinel-2 satellite images for monitoring the rice transplanting date, harvest date, and number of rice croppings. Its result may reveal the rice phenological dates for three croppings in monsoon Asia. However, three major points needed to be improved.*

We greatly appreciate the valuable and constructive comments provided by the reviewer. In the following responses, we will adress each comment point-by-point. The responses to the comments are presented in blue font, and changes to the manuscript are highlighted in red with a grey background.

*#1 The authors use two steps to map the rice calendar: (1) detection of rice phenological dates and number of rice croppings through combination of a feature-based algorithm and the fitted Weibull function, and (2) spatio-temporal integration of the detected transplanting and harvest dates using von Mises maximum likelihood estimates. However, there is no logical relationship between the two steps. In its current version, it seems that the authors use two different methods (combination of a feature-based algorithm and the fitted Weibull function, and von Mises maximum likelihood estimates) to monitor the rice phenological dates, but did not compare the advantages and disadvantages of the two methods. Maybe the output from step one would be the input to step two? The authors should explain the logical relationship, because this is the key of this study.*

**Response:** We extend our sincere appreciation to the reviewer for the valuable feedback. As you rightly pointed out, the output from step one would serve as the input for step two. In Step 1, we utilized a feature-based algorithm to extract the transplanting and harvest date (as detailed described in Lines 181-205). To identify the number of rice croppings, a fitted Weibull function was employed (as detailed described in Lines 207-226). Upon completing Step 1, we successfully detected all the transplanting and harvest dates across two years for each grid. However, it is important to note that these detected transplanting and harvest dates within each grid can vary annually due to different weather conditions, the impact of climate change, adjustments in agricultural schedules, and the availability of satellite images. Furthermore, the transplanting and harvest dates for a specific cropping season in one grid can significantly differ from those in neighboring grids, possibly indicating detection errors. Therefore, the spatio-temporal integration of these detected transplanting and harvest dates, referred to as Step 2, is a necessary process for generating a multi-year spatially averaged rice calendar. In response to your suggestion, we have revised the 2.3.2 Section to emphasize the key points that Step 1 is the input of Step 2. The revised contents are as follows:

➤ All the transplanting and harvest dates across two years for each grid were detected in Step 1 by using the algorithms and processes described above. However, these detected transplanting and harvest dates in each grid vary annually due to different weather conditions, the effects of climate change, adjustments in agricultural schedule, and the availability of satellite images. Additionally, the detected transplanting and harvest dates for a specific cropping season in a grid can differ markedly from those in neighboring grids, possibly indicating detection errors. Therefore, the temporal and spatial integration of the detected transplanting and harvest dates, referred to as Step 2, is a necessary step for generation of a multi-year, spatially averaged rice calendar (Lines 237-242).

We have made a revision that makes the logical relationship between Step 1 and Step 2 more clear in the Abstract:

➢ The methodological framework incorporates two steps: (1) detection of rice phenological dates and number of rice croppings through combination of a feature-based algorithm and the fitted Weibull function, and (2) spatio-temporal integration of the detected transplanting and harvest dates derived from step 1 using von Mises maximum likelihood estimates (Lines 18-21).

At the same time, we have added arrows in Figure 2 to show that the transplanting and harvest dates obtained in Step 1 are the inputs for Step 2. Correspondingly, its description and caption have also been revised as follows:

➢ The overall methodology for rice calendar mapping, which is summarized in Fig. 2, can be divided into two steps. The first step is extraction of transplanting and harvest dates and detection of the number of rice croppings, depicted in Fig. 2 Step 1. 1 as the algorithm for phenological dates and number of rice croppings detection and in Fig. 2 Step 1-2 as the process of phenological dates and number of rice croppings detection. The transplanting and harvest dates obtained in the first step (Step 1) require temporal and spatial integration for the generation of the rice calendar (Fig. 2 Step 2). The following sections provide elaboration on the major procedures involved in each step (Lines 168-173).

[Figure]

**Figure 2** Workflow for gridded rice calendar mapping based on satellite images. Step 1 depicts the algorithm and process of transplanting and harvest dates extraction, along with the detection of number of rice croppings, as shown in the first box. In Step 2, the generation of the rice calendar is described, relying on the detected transplanting and harvest dates derived from Step 1, through the temporal and spatial integration of the detected phenological dates displayed in the second box (Page 8).

Moreover, we have improved the paragraph that emphasizes the derivation of Step 2 from Step 1 in the Results and Discussion section:

➢ Temporal and spatial integration of detected transplanting and harvest dates, as derived from Step 1, pose a great challenge owing to flexible agricultural schedules, and the availability of satellite imagery (Lines 454-455).

*#2 The validation of the accuracy of the results is weak and scarce. The authors only compare the identified rice phenological dates with the existing products, including the RiceAtlas rice calendar, the RICA rice calendar, and the SAGE rice calendar, instead of field observation of rice phenology. It is inappropriate because the existing products themselves have identification errors, thus lacking of reliability. Therefore, it is meaningless to verify the detection results using the biased information. The authors should add the contents about verifying the results with actual field observations of rice phenology.*

**Response:** We sincerely thank the reviewer for the valuable comment. We acknowledge the critical role of rice calendar accuracy in validating the precision of our proposed rice calendar. We completely understand the review's concerns about the accuracy of existing rice calendars, which may contain the uncertainties due to several reasons. It is worth noting that we have addressed this issue in detail in the "3.4 Accuracy of reference rice calendars" section of the original version, as follows:

"*Accuracy of reference rice calendar*. Although the proposed rice calendar showed reasonable performance in comparison with that of the other rice calendars, it should consider the accuracy of the reference rice calendars. In particular, when evaluating transplanting and harvest dates against the RiceAtlas rice calendar, it is worth noting that the phenological dates were sources from census data, database, publications, and report compilations (Laborte et al., 2017). Another concern regarding the RiceAtlas rice calendar is the overlay of phenological dates between rice cropping seasons. Additionally, the RiceAtlas, RICA, and SAGE rice calendars are based on the administrative scale, resulting in large spatial coverage with only one recorded

phenological date and number of rice croppings (Fig. S5-S7). It should also be mentioned that some rice calendars, e.g., SAGE, are poorly documented and do not record triple croppings (Scaks et al., 2010) (Lines 494-501)."

Regarding RiceAtlas, despite the presence of detection errors, it remains the only detailed global rice calendar to date (Laborte et al., 2017). Since its publication in 2017, RiceAtlas rice calendar has been cited 84 times (according to Web of Science). It stands as the widely accepted rice calendar extensively used in numerous research domains, including rice calendar validation (e.g., RICA rice calendar in Mishra et al. (2021), Iizumi et al. (2019)), mapping rice paddy field distribution (Han et al., 2021; Luintel, et al., 2021), predicting rice production (Oort et al., 2017; Wu et al., 2023), and estimating methane emissions (Crippa et al., 2020; Ouyang et al., 2023). In fact, researchers regard RiceAtlas as the standard database for comparing rice phenology based on earth observation (Mishra et al., 2021).

Our primary objective here is to compare the differences among similar large-scale rice calendar products. The improved accuracy of our proposed rice calendar, which outperformed the RICA rice calendar when compared with RiceAtlas rice calendar, highlights the critical importance and necessity of our proposed rice calendar. Our proposed rice calendar can integrate the strengths, such as consistent detection through remote sensing methods as seen in the RICA rice calendar, while addressing the weaknesses, such as the coarse spatial resolution found in existing large-scale rice calendars like RiceAtlas and RICA rice calendars and limited number of rice cropping seasons in the SAGE rice calendar). This effort has resulted in the development of our new, relatively finer rice calendar.

As reviewer rightly point out, actual field observations are essential for validating the accuracy of rice phenology detection methods. Fortunately, we have previously validated the rice phenology detection methods (feature-based algorithm) used in this rice calendar production in a prior study using actual field observations (Zhao et al., 2023). Results revealed a bias of 4 and -13 days for transplanting and harvest dates,

respectively (as shown in Fig. 6, Fig. 7, Table 1, and Appendix A in Zhao et al. (2023)). Moreover, we employed various spatial scales, including actual site observations, sub-national, and 0.5° gridcell scales, to rigorously validate our feature-based algorithm, ensuring its robustness (Zhao et al., 2023). We have added this description in the revised manuscript as follows:

➢ A feature-based algorithm has been proposed for rice phenology detection at the large scale (Zhao et al., 2023). The superiority of this algorithm lies in the use of backscattering (VH) and vegetation indices (Enhanced Vegetation Index (EVI) and Normalized Yellow Index (NDYI)) derived from Sentinel-1 and Sentinel-2 images, which reflect features related to rice cultivation such as flooding, maximum leaf area, and most yellowness around transplanting, heading, and harvest date. Additionally, this algorithm has successfully tracked rice phenological dates in different cropping systems (single, double, and triple croppings) and at different spatial scales (sub-nation, 0.5° gridcell, and site scales) (Zhao et al., 2023). Thus, the recognition of rice phenological dates is based on the feature-based algorithm (Lines 79-85).

➢ The feature-based algorithm was used on the smoothed VH, EVI, and NDYI time series data to capture the aforementioned phenological characteristics of rice crops (Zhao et al., 2023). This algorithm's robustness has been confirmed at multiple spatial scales (sub-nation, 0.5° gridcell, and site scales) and cropping systems (single, double, and triple croppings) in monsoon Asia (Zhao et al., 2023). The transplanting date was determined by identifying the minimum VH intensity from the shortest plants above the water surface. As the rice plants grow above the water surface and interact with the incident radar signal, the VH intensity gradually increases (Torres et al., 2012). The harvest date was detected using the yellow signal derived from the NDYI, which employs a combination of green and blue bands to represent the balance between rice growth and senescence. Consequently, the NDYI value reaches a peak (approaches nearly 0 from negative values),

indicating the maximum yellowness associated with the harvest date (Zhao et al., 2023) (Fig. 2 Step 1 Algorithm a) (Lines 187-193).

In summary, our proposed rice calendar has undergone validation using other rice calendar products as well as through field observations of rice phenology. We hope this addresses your concerns adequately.

**References**

Zhao, X., Nishina, K., Akitsu, T.K., Jiang, L., Masutomi, Y., Nasahara, K.N.: Feature-based algorithm for large-scale rice phenology detection based on satellite images, Agric. For. Meteorol., 329, 109283, https://doi.org/10.1016/j.agrformet.2022.109283, 2023.

Laborte, A.G., Gutierrez, M.A., Balanza, J.G., Saito, K., Zwart, S.J., Boschetti, M., Murty, M.V.R., Villano, L., Aunario, J.K., Reinke, R., Koo, J., Hijmans, R.J., and Nelson, A.: RiceAtlas, a spatial database of global rice calendars and production, Sci. Data, 4, 170074, https://doi.org/10.1038/sdata.2017.74, 2017.

Mishra. B., Busetto, L., Boschetti, M., Laborte, A., Nelson, A.: RICA: A rice crop calendar for Asia based on MODIS multi year data. Int. J. Appl. Earth Obs. Geoinf. 103, 102471, https://doi.org/10.1016/j.jag.2021.102471, 2021.

Iizumi, T., Kim, W., and Nishimori, M.: Modeling the global sowing and harvesting windows of major crops around the year 2000. J. Adv. Model. Earth Syst., 11(1), 99-112, https://doi.org/10.1029/2018MS001477, 2019.

Han, J., Zhang, Z., Luo, Y., Cao., J., Zhang, L., Cheng, F., Zhuang, H., Zhang, J., Tao, F.: NESEA-Rice10: high-resolution annual paddy rice maps for Northeast and Southeast Asia from 2017 to 2019. Earth Syst. Sci. Data, 13, 5969-5986, https://doi.org/10.5194/essd-13-5969-2021, 2021.

Luintel, N., Ma, W., Ma, Y., Wang, B., Xu, J., Dawadi, B., Mishra, B.: Tracking the dynamics of paddy rice cultivation practice through MODIS time series and PhenoRice algorithm. Agric. For. Meteorol., 307, 108538, https://doi.org/10.1016/j.agrformet.2021.108538, 2021.

van Oort, P.A.J. and Zwart, S.J.: Impacts of climate change on rice production in Africa and causes of simulated yield changes. Glob. Change Biol., 24, 1029-1045, https://doi.org/10.1111/gcb.13967, 2018.

Wu, H., Zhang, J., Zhang, Z., Han, J., Cao, J., Zhang, L., Luo, Y., Mei, Q., Xu, J., Tao, F.: AsiaRiceYield4km: seasonal rice yield in Asia from 1995 to 2015. Earth Syst. Sci. Data, 15, 791-803, https://doi.org/10.5194/essd-15-791-2023, 2023.

Crippa, M., Solazzo, E., Huang, G., Guizzardi, D., Koffi, E., Muntean, M., Schieberle, C., Friedrich, R., Janssens-Maenhout, G.: High resolution temporal profiles in the emissions database for global atmospheric research. Sci. Data, 7, 121, https://doi.org/10.1038/s41597-020-0462-2, 2020.

Ouyang, Z., Jackson, R.B., McNicol, G., Fluet-Chouinard, E., Runkle, B.R.K., Papale, D., Knox, S.H., Cooley, S., Delwiche, K.B., Feron, S., Irvin, J.A., Malhotra, A., Muddasir, M., Sabbatini, S., Alberto, M.C.R., Cescatti, A., Chen, C., Dong, J., Fong, B.N., Guo, H., Hao, L., Iwata, H., Jia, Q., Ju, W., Kang, M., Li, H., Kim, J., Reba, M.L., Nayak, A.K., Roberti, D.R., Ryu, Y., Swain, C.K.,Tsuang, B., Xiao, X., Yuan, W., Zhang, G., Zhang, Y., Paddy rice methane emissions across Monsoon Asia. Remote Sens. Environ., 194, 348-365, 284, 113335, https://doi.org/10.1016/j.rse.2022.113335, 2023.

*#3 The authors emphasize for many times that the proposed rice calendar fills the gaps in high resolution rice calendars, like Line 27, and Line 96 (high spatial resolution, 0.5°). However, there are currently so many high resolution satellites images, like Landsat images for 30 m and Sentinel-1/2 satellite images for 10 m. The spatial resolution for 0.5° of the proposed rice calendar cannot be called high resolution.*

**Response:** We appreciate the constructive feedback provided by the reviewer and have incorporated this suggestion into revised manuscript. As you noted, it is indeed true that our proposed rice calendar, with a spatial resolution of 0.5°, may not be categorized as "high resolution" when compared to the 30 m or even 10 m high-resolution satellite images. It is also important to convey the advantage of our proposed rice calendar – our spatial resolution surpasses that of existing large-scale rice calendars, which are typically produced at administrative scale ranging from country to sub-country levels. Taking into account these two key aspects, we have refrained from using the term "high

spatial resolution (0.5°)" and have also decided to quit using "finer-resolution rice calendar" in the revised manuscript.

The revised contents are specifically as follows:

➢ This novel gridded rice calendar fills the gaps in half-degree rice calendars across major global rice production areas, facilitating research on rice phenology that is relevant to the climate change (Lines 27-28).

➢ The objective of this study was to develop a new gridded rice calendar that highlights the following features: (a) consistent detection using remote-sensing methods, (b) spatial resolution (0.5° × 0.5°), (c) large-scale coverage (monsoon Asia), and (d) ability to extract multiple rice croppings (Lines 95-97).

➢ The advantages of the above-mentioned algorithms (Fig 2 Step 1, Step 2) largely contribute to the production of a gridded rice calendar.  The proposed rice calendar provides spatially explicit rice phenology with continental coverage through remote sensing methods. The major difference between our proposed rice calendar and the RICA rice calendar lies in the use of a feature-based algorithm with VH and NDYI, which allows our proposed rice calendar to theoretically estimate rice phenology more accurately. Zhao et al. (2023) demonstrated that VH can accurately capture the start of paddy water logging, and NDYI is a good indicator of rice maturity stage. Our proposed rice calendar presents a highly patchy map of rice phenological information (Figs. 6 and 10a). The 0.5° resolution of our proposed rice calendar is finer than that of other rice calendars, including RiceAtlas, RICA at sub-national scale, and SAGE derived from sub-national data. This improvement greatly reduces the bias error caused by assigning averaged rice phenology to administrative units, as rice phenology can vary considerably within large administrative units (Franch et al., 2022). Furthermore, our proposed rice calendar displays the detailed distribution of rice paddy fields (Figs. 6 and 10a), in contrast to previous rice calendars that

covered entire administrative areas, irrespective of the small proportion of rice cultivation (Figs. S5-S7 and 10b-d) (Lines 466-474).

➤ Given the absence of an updated global/continental-scale rice calendar that can explicitly depict spatial gridded transplanting date and harvest date information, and the number of rice croppings, this study developed a new gridded rice calendar for monsoon Asia with spatially explicit detail of rice phenology using a new methodological framework based on Sentinel-1 and Sentinel-2 images (Lines 526-529).

*Minor issues*

1. *Line 129-130, the authors aggregated rice distribution map at 500 m resolution into 0.5° resolution by randomly selecting 20 rice fields to derive the average phenology. This process is unreasonable because 20 fields are not representative. It is suggested that the authors should first calculate the planting fraction for rice paddy at 0.5° resolution based on rice distribution map at 500 m resolution, derive the pixels which have a higher planting proportion like 80%, and then extracting the average phenology of above pixels.*

**Response:** We greatly appreciate the reviewer's constructive comments. Our apologies for the ambiguous description in the previous version. The rice paddy filed distribution map was aggregated to a gridded map with 0.5° resolution, INSTEAD OF converting the rice paddy field distribution map at 500 m resolution into 0.5° resolution by randomly selecting 20 rice fields to derive the average phenology. To avoid ambiguity, we have moved the sentence "This rice paddy field distribution map was aggregated to a gridded map with 0.5° resolution (Fig. 1b)." (originally located at the beginning of the sampling method paragraph) to the end of the paragraph describing the rice paddy field distribution map.

➢ The rice paddy field distribution map adopted in this study is from a 500 m resolution map produced using MODIS images (Zhang et al., 2020) (Fig. 1a). This map effectively displays the presence and distribution of rice paddy fields over monsoon Asia. The reliability of this map is substantiated by its strong correlation with existing rice paddy field maps across diverse areas ($R^2$ values ranging from 0.72 to 0.95) and its alignment with the area information obtained from FAOSTAT statistical data for each country (Zhang et al., 2021).  In this study, this rice paddy field distribution map was aggregated into a gridded map with 0.5° resolution (Fig. 1b).

 Within each 0.5° grid, 20 rice paddy fields were randomly selected to derive the average rice phenology for that grid (Xiao et al., 2021; Zhao et al., 2023). This sampling method effectively minimizes errors caused by misclassification of rice paddy fields by excluding outliers that deviate from the averaged rice phenology (Zhao et al., 2023) (Lines 121-132).

In fact, we have already calculated the fraction of rice paddy fields at 0.5° resolution based on the rice distribution map at 500 m resolution in our research. Fig. 1b displays the percentage of rice paddy field in 0.5° grids. Green gradient indicates variation in the percentage coverage of rice paddy fields (Page 5).

From the provided fraction of the rice paddy field map (Fig. 1b), areas with a high proportion of rice cultivation are mainly located in the Indo-Gangetic Plain, the Yangtze Plain, the Ayeyarwady Delta region, and the Mekong Basin. If we extract the average phenology from these high proportion grids, we will miss the rice phenology in other grids across monsoon Asia. However, the objective of our research is to develop a gridded rice calendar that considers each grid individually. Indeed, the high proportion of rice paddy areas improves the accuracy of averaged rice phenology detection by enabling more precise identification of rice paddy fields. In contrast, our sampling

method – randomly selecting 20 rice paddy fields – is suitable for both high and low proportion rice paddy area grids. It effectively selects the rice paddy fields, saving computation time and facilitates the implementation, while reducing the error of misclassification of rice paddy fields by excluding outliers that deviate from the averaged rice phenology (Fig. S2 in Supplementary in Zhao et al., 2023). The average rice phenology for each grid was therefore determined by the rice phenology that predominates in that grid.

[Figure]

**Figure 1** Location of the study area and distribution of rice paddy fields in monsoon Asia. Rice paddy field distribution map (**a**) was obtained from Zhang et al., (2020), which was produced using MODIS images. Green areas indicate rice paddy fields. Gridded rice paddy field map (**b**) shows the percentage of rice paddy field in 0.5° grids. Green gradient indicates variation in the percentage coverage of rice paddy fields (Page 5).

2. *Line 175-178, Step 1-1 and Step 1-2 describe the same thing. It is suggested that these two steps be combined.*

**Response:** We greatly appreciate the reviewer's suggestion. As you suggested, Step 1-1 and Step 1-2 depict the algorithms and processes for detecting rice phenological dates and number of rice croppings, respectively. Both of these sub-steps focus on the same issue: how to detect phenological dates and the number of rice croppings. Therefore, we have combined Step 1-1 and Step 1-2 into a comprehensive step, summarizing these two sub-steps as "Step 1 Detection of transplanting and harvest dates, number of rice croppings". Additionally, we have revised Figure 2 and its description to correspond to this combination.

➤ The overall methodology for rice calendar mapping, which is summarized in Fig. 2, can be divided into two steps. The first step is extraction of transplanting and harvest dates and detection of the number of rice croppings, depicted in Fig. 2 Step 1.  The transplanting and harvest dates obtained in the first step (Step 1) require temporal and spatial integration for the generation of the rice calendar (Fig. 2 Step 2). The following sections provide elaboration on the major procedures involved in each step (Lines 168-173).

[Figure]

**Figure 2** Workflow for gridded rice calendar mapping based on satellite images. Step 1 depicts the algorithm and process of transplanting and harvest dates extraction, along with the detection of number of rice croppings, as shown in the first box. In Step 2, the generation of the rice calendar is described, relying on the detected transplanting and harvest dates derived from Step 1, through the temporal and spatial integration of the detected phenological dates displayed in the second box (Page 8).

We have also revised the description of Step 1-1 and Step 1-2 in other contents as follows:

➢ **2.3.1.1 Algorithm  and process  for extraction of transplanting and harvest dates** (Line 180).

➢ Additionally, the time of rice harvest is characterized by irreversible yellowing of the leaves, resulting from the rapid breakdown of chlorophyll and the

photosynthetic apparatus (Zhang et al., 2021b) (Fig. 2 Step 1 Algorithm) (Lines 184-185).

➤ Consequently, the NDYI value reaches a peak (approaches nearly 0 from negative values), indicating the maximum yellowness associated with the harvest date (Zhao et al., 2023) (Fig. 2 Step 1 Algorithm a) (Lines 192-193).

➤ The minimum VH and peak NDYI were detected within the time window (Fig. 2 Step 1 Process a), indicating that only the minimum VH and the maximum NDYI values within the time window, before and after the EVI peak, can be identified as the transplanting date and harvest date, respectively (Lines 195-197).

➤ If the peak NDYI could not be obtained from those time windows, peak NDYI was identified using the peak EVI date ($DOY_{EVI_{max}}$) plus the corresponding difference days for each rice cropping, as referenced in Zhao et al. (2013) (Fig. 2 Step 1 Process a) (Lines 203-205).

➤ **2.3.1.2 Method  and process  for detecting the number of rice croppings** (Line 206).

➤ The six-parametric Weibull function, $w(x) = \left(d + \exp\left(-\left(\frac{x}{e}\right)^f\right)\right) \times \left(1 - a\exp\left(-\left(\frac{x}{b}\right)^c\right)\right)$ (where $a$, $b$, $c$, $d$, $e$, and $f$ are the free parameters to be fitted) (Rolinski et al., 2007), can be used to identify the number of rice croppings by depicting an arc with the shape of downward-opening patterns from the smoothed EVI time series (Fig. 2 Step 1 Algorithm b) (Lines 207-209).

➤ After application of the function (Eq. (7)), all available arcs of the smoothed EVI time series were then labelled, including the start (start day of detected EVI arc, $DOY_{EVI\ arc_{first\ day}}$), peak (peak day of detected EVI arc, $DOY_{EVI_{max}}$), and end (end day of detected EVI arc, $DOY_{EVI\ arc_{last\ day}}$) of the arc, and the peak EVI value ($Value_{EVI_{max}}$) (Fig. 2 Step 1 Process b) (Lines 219-221).

➤ A rice feature-based phenology algorithm (Zhao et al., 2023) was applied to Sentinel-1 and Sentinel-2 images to detect gridded rice transplanting and harvest dates (Fig. 2 Step 1 Algorithm a) (Lines 432-433).

➢ The fitted Weibull function, implemented in the R package (Fig. 2 Step 1 Algorithm b), automatically detects the number of rice croppings based on the shape of the smoothed EVI time series, facilitating rapid and efficient rice calendar mapping (Lines 447-448).

---

## Author Comment (AC2)

**Response to the anonymous reviewer's comments**

*Anonymous referee #2:*

*In my opinion, the study is original and such datasets are needed; also, the approaches implementing Sentinel data in these tasks are important to develop. However, I have some major concerns regarding the manuscript which are described below.*

We greatly appreciate the valuable and constructive comments provided by the reviewer. In the following responses, we will adress each comment point-by-point. The responses to the comments are presented in blue font, and changes to the manuscript are highlighted in red with a grey background.

*In the abstract, the authors state that other datasets are characterized by coarse resolution, while their dataset is, in fact, very coarse (0.5 degrees, approximately 55km). I suggest rephrasing or clarifying this statement. Also, why is the final calendar characterized by such low resolution when Sentinel data used in this study have a resolution of 10m?*

**Response:** We sincerely appreciate the reviewer's comments, and we acknowledge that the description of our rice calendar dataset may not be suitable. Although we want to convey the advantage of our proposed rice calendar – our spatial resolution surpasses that of existing large-scale rice calendars, which are typically produced at administrative scale ranging from country to sub-country levels – we have refrained from using the term "finer-resolution" in the Abstract section and "high spatial resolution" in the main text.

The revised contents are specifically as follows:

➢ This novel gridded rice calendar fills the gaps in half-degree rice calendars across major global rice production areas, facilitating research on rice phenology that is relevant to the climate change (Lines 27-29).

➢ The objective of this study was to develop a new gridded rice calendar that highlights the following features: (a) consistent detection using remote-sensing methods, (b) spatial resolution (0.5° × 0.5°), (c) large-scale coverage (monsoon Asia), and (d) ability to extract multiple rice croppings (Lines 110-112).

➢ The advantages of the above-mentioned algorithms (Fig 2 Step 1, Step 2) largely contribute to the production of a gridded rice calendar.  The proposed rice calendar provides spatially explicit rice phenology with continental coverage through remote sensing methods. The major difference between the proposed rice calendar and the RICA rice calendar lies in the use of a feature-based algorithm with VH and NDYI, which allows the proposed rice calendar to theoretically estimate rice phenology more accurately. Zhao et al. (2023) demonstrated that VH can accurately capture the start of paddy water logging, and NDYI is a good indicator of rice maturity stage. The proposed rice calendar presents a highly patchy map of rice phenological information (Figs. 6 and 10a). The 0.5° resolution of the proposed rice calendar is finer than that of other rice calendars, including RiceAtlas, RICA at sub-national scale, and SAGE derived from sub-national data. This improvement greatly reduces the bias error caused by assigning averaged rice phenology to administrative units, as rice phenology can vary considerably within large administrative units (Franch et al., 2022). Furthermore, the proposed rice calendar displays the detailed distribution of rice paddy fields (Figs. 6 and 10a), in contrast to previous rice calendars that covered entire administrative areas, irrespective of the small proportion of rice cultivation (Figs. S6-S8 and 10b-d) (Lines 501-513).

➢ Given the absence of an updated global/continental-scale rice calendar that can explicitly depict spatial gridded transplanting date and harvest date information, and the number of rice croppings, this study developed a new gridded rice calendar for monsoon Asia with spatially explicit detail of rice phenology using a new methodological framework based on Sentinel-1 and Sentinel-2 images (Lines 586-589).

The utilization of Sentinel images with a 10 m resolution contributes to improving the accuracy of rice calendar mapping because some rice paddy fields tend to be smaller than the 500 m resolution (Mishra et al., 2021). Sentinel satellites provide dense and detailed observations that can overcome the longer temporal frequency of Landsat images and the spatial resolution limitations of MODIS in mapping rice calendars. The objective of this work is to develop a large-scale rice calendar for monsoon Asia based on a more versatile methodology framework. The production of proposed half-grid rice calendar from Sentinel images is based on the following considerations: First, it greatly reduces the computation time and efficiently provides rice phenology information at a large scale. Second, the spatial resolution of proposed rice calendar surpasses that of existing large-scale rice calendars. Third, the 0.5° resolution fulfills the requirements for model simulation in estimating greenhouse gas emissions estimation.

*Furthermore, I have concerns regarding the accuracy assessment of the calendar. The dates of transplanting and harvesting are validated with the Rice Atlas, which has a national or subnational resolution, correct? It seems, therefore, inappropriate to validate the proposed calendar using such data.*

**Response:** We appreciate the reviewer's constructive comments. In our prior study (Zhao et al., 2023), we have already conducted rigorous validation of the rice phenology detection methods (feature-based algorithm) used in this calendar production at multiple spatial scales (sub-nation, 0.5° gridcell, and site scales) and across various cropping systems (single, double, and triple croppings) in monsoon Asia. Results revealed biases in transplanting dates of 2, 0, and 4 days, while that in harvest dates of 2, -5, and -13 days at the sub-nation, 0.5° gridcell, and site scales, respectively (as demonstrated in Fig. 6, Fig. 7, Table 1, and Appendix A in Zhao et al. (2023)). This rigorous validation underscores the robustness of our proposed rice calendar (Zhao et al., 2023). Your point made us realize that we did not mention enough in the first draft about the results of past validation. So, we have added an explanation for this in the revised manuscript.

For the reason that we have conducted detailed validation in the previous study (Zhao et al., 2023), we have limited our dataset comparison in this paper to only those with broad datasets covering only the same geographic areas. Despite the coarse spatial resolution that does not correspond to our proposed rice calendar, RiceAtlas remains the only detailed global rice calendar to date (Laborte et al., 2017). Since its publication in 2017, RiceAtlas rice calendar has been cited 84 times (according to Web of Science). It stands as the widely accepted rice calendar extensively used in numerous research domains, including rice calendar validation (e.g., RICA rice calendar in Mishra et al. (2021) and model-based rice calendar in Iizumi et al. (2019)), mapping rice paddy field distribution (Han et al., 2021; Luintel et al., 2021), predicting rice production (Oort et al., 2017; Wu et al., 2023), and estimating methane emissions (Crippa et al., 2020; Ouyang et al., 2023). In fact, researchers regard RiceAtlas as the standard database for comparing rice phenology based on earth observation (Mishra et al., 2021). Therefore, we employed the RiceAtlas rice calendar to validate the transplanting and harvest dates. Our primary objective here is to compare the differences among similar large-scale rice calendar products. The absence of a half-grid rice calendar highlights the critical

importance and necessity of our proposed rice calendar. We have included these points in the revised manuscript as follows:

➢ A feature-based algorithm, proposed for large-scale rice phenology detection (Zhao et al., 2023), excels in utilizing backscattering (VH) and vegetation indices (Enhanced Vegetation Index (EVI) and Normalized Yellow Index (NDYI)) derived from Sentinel-1 &-2 to reflect features related to rice cultivation such as flooding, maximum leaf area, and most yellowness around transplanting, heading, and harvest date. Additionally, this algorithm has successfully tracked rice phenological dates in different cropping systems (single, double, and triple croppings) and at different spatial scales (sub-nation, 0.5° gridcell, and site scales) (Zhao et al., 2023) (Lines 94-99).

➢ These phenological characteristics of rice crops can be captured by a feature-based algorithm applied on smoothed VH, EVI, and NDYI time series data (Zhao et al., 2023). This algorithm's robustness has been confirmed at multiple spatial scales (sub-nation, 0.5° gridcell, and site scales) and cropping systems (single, double, and triple croppings) in monsoon Asia (Zhao et al., 2023). The transplanting date was determined by identifying the minimum VH intensity from the shortest plants above the water surface, where VH intensity gradually increase as they interact with the radar signal (Torres et al., 2012). The harvest date was detected using the NDYI's yellow signal, indicating the maximum yellowness at harvest date (Zhao et al., 2023) (Fig. 2 Step 1 Algorithm a) (Lines 210-221).

■ *In the abstract and in the whole manuscript I suggest changing units of area from 5.3 × 10⁶ to millions of kilometres, for example 5.3 mln of km.*

**Response:** Our apologies. The correct area is $5.3 \times 10^5$ km$^2$ instead of $5.3 \times 10^6$ km$^2$ according to the y-axis of Figure 11. Following your suggestion, we have converted the unit of area from $5.3 \times 10^5$ km$^2$ to 0.53 million of km$^2$. The revised contents are as follows:

➢ In total, the proposed rice calendar can detect single, double, and triple rice cropping with area of 0.53, 0.45, and 0.09 million of km$^2$, respectively (Lines 25-27).

➢ The areas covered by single, double, and triple rice croppings in the proposed rice calendar area 0.53, 0.45, and 0.09 million of km$^2$, respectively (Fig.11). The area covered by single rice cropping falls within the range of 0.24 million of km$^2$ (RiceAtlas rice calendar) to 0.65 million of km$^2$ (RICA rice calendar) (Fig.11) (Lines 416-418).

➢ Additionally, the area covered by triple rice cropping falls within the range of 0.05 million of km$^2$ (RICA rice calendar) to 0.4 million of km$^2$ (RiceAtlas rice calendar)

■ *Line 106: rephrase to something like "The analysed area.... (Fig 1)"*

**Response:** Thanks for your suggestions. We have rephrased the description of study area. The revised sentence is as follows:

➤ The analysed area is located in monsoon Asia, which covered the region of 10° S to 53.5° N, 61° E to 153° E (Lines 121-122).

■ *Figure 1 – so the study area includes all the countries shown in grey, with bold black borders – it is not fully clear from the figure, maybe add to legend. Also consider adding country names to the map.*

**Response:** We appreciate the reviewer's suggestion. We have added the legend of bold black borders to indicate the countries in the study area. At the same time, we have added the country names to the map. The revised Figure 1 is shown as follows:

[Figure]

**Figure 1.** Location of the study area and distribution of rice paddy fields in monsoon Asia. Rice paddy field distribution map **(a)** was obtained from Zhang et al., (2020), which was proposed using MODIS images. Green areas indicate paddy fields, and bold black borders indicate the countries in this study area. Gridded rice paddy field map (b) shows the percentage of rice paddy field in 0.5° grids. Green gradient indicates variation in the percentage coverage of rice paddy fields (Page 5).

- *Line 130 – what is the average size of a paddy field in these regions, is 20 samples enough to represent it for such large grid?*

**Response:** Thank you for your question. The average size of paddy fields in a grid is 180 km². This area was calculated by the percentage of rice paddy field in each grid (shown as Fig. 1b) multiplying the area of each grid cell on the ellipsoidal earth (Fig. S9). The distribution of the area of rice paddy fields in monsoon Asia is shown in the following figure, where 75% of grids have paddy rice filed areas of less than 180 km². Thus, one sample could represent a rice paddy field of 9 km², equivalent to 3,000 square meters. Although there is a slight gap between this and exact site-scale rice phenology, but it efficiently saves computation time and facilitates the implementation to obtain such broad-scale averaged rice phenology, based on the assumption that rice phenology is stable within a specific area.

[Figure]

Moreover, to demonstrate the rationale for sampling 20 rice paddy fields, we provide an example to address this issue, as shown in the following figure. In this example, 10,

20, 30, 40, and 50 rice paddy fields were randomly selected within four neighboring 0.5° grids. There is no difference in the extracted EVI peak days among the 20, 30, 40, and 50 sampled rice paddy fields for four grids. However, the 10 sampled rice paddy fields tend to yield different peak EVI days. We have included this figure as Fig. S2 in the Supplementary.

[Figure]

**Figure S2.** Example of the effect of sampling numbers in rice paddy fields (10, 20, 30, 40, and 50) for extracting phenological dates. X-axis denotes the days from 1 January 2019 to 31 December 2020, and Y-axis denotes the EVI values. The red line denotes the smoothed EVI values. The number in each panel denotes the extracted peak EVI days.

We have cited Fig. S2 in the "Rice paddy field distribution map and sampling method"

section to show how the sample size was determined, as follows:

➢ Additionally, this sampling size of 20 rice paddy fields is sufficiently enough that saves computation time and has no effect on averaged rice phenology detection (Fig. S2) (Lines 150-151).

■ *Line 135 – the codes/notebooks should be included in the paper.*

**Response:** Thanks for your comments. We have added this code in the "Code availability" section as follows:

➢ The code for getting VH/EVI/NDYI time series data from Sentinel-1 and Sentinel-2 images, extracting the transplanting and harvest dates from smoothed VH/EVI/NDYI time series data, and spatial and temporal integration of detected transplanting and harvest dates can be found at https://db-test.cger.nies.go.jp/DL/10.17595/20230728.001.html.en (Zhao and Nishina, 2023) (Lines 580-583).

■ *Line 160 – these are widely used metrics; I think their equations are redundant*

**Response:** We agree with you that these equations are widely used. Consequently, we have moved them to Supplementary Text 1. The revised content in the Manuscript is as follows:

➢ The RiceAtlas rice calendar, with its detailed phenological date range, was used to assess the performance of the proposed rice calendar in determining  transplanting date and harvest date, evaluating it based on

  the coefficient of determination ($R^2$), bias error (Bias), Mean Absolute Error (MAE), and Root Mean Square Error (RMSE) (Supplementary Text 1)  (Lines 175-186).

$$R^2 = 1 - \frac{(\sum_{i=1}^{n}(y_i-\bar{y})(s_i-\bar{s}))^2}{\sum_{i=1}^{n}(y_i-\bar{y})^2-\sum_{i=1}^{n}(s_i-\bar{s})^2} \tag{3}$$

$$Bias = \frac{1}{n}\sum_{i=1}^{n}(y_i - s_i) \tag{4}$$

$$MAE = \frac{1}{n}\sum_{i=1}^{n}|y_i - s_i| \tag{5}$$

$$RMSE = \sqrt{\frac{1}{n}\sum_{i=1}^{n}(y_i - s_i)^2} \tag{6}$$

The revised content in the Supplementary Text 1 is as follows:

➢ **1. Performance of the proposed rice calendar**

The performance of the proposed rice calendar in determining the transplanting and harvest dates was assessed using the coefficient of determination ($R^2$), bias error (Bias), Mean Absolute Error (MAE), and Root Mean Square Error (RMSE) which were calculated as follows:

$$R^2 = 1 - \frac{(\sum_{i=1}^{n}(y_i-\bar{y})(s_i-\bar{s}))^2}{\sum_{i=1}^{n}(y_i-\bar{y})^2-\sum_{i=1}^{n}(s_i-\bar{s})^2} \tag{1}$$

$$Bias = \frac{1}{n}\sum_{i=1}^{n}(y_i - s_i) \tag{2}$$

$$MAE = \frac{1}{n}\sum_{i=1}^{n}|y_i - s_i| \tag{3}$$

$$RMSE = \sqrt{\frac{1}{n}\sum_{i=1}^{n}(y_i - s_i)^2} \tag{4}$$

where $y_i$ and $\bar{y}$ are the phenological dates from the proposed rice calendar for sample grid ($i$) and the corresponding mean value, respectively, $s_i$ and $\bar{s}$ are the phenological dates from the reference rice calendar for sample grid ($i$) and the corresponding mean value, respectively, and $n$ represents the number of sampled phenological dates.

- *Lines 197-205 – this is very complex and somehow hard to follow. Maybe it can be put into a table and moved it to supplementary materials. Also, why such dates were used?*

**Response:** We greatly appreciate the reviewer's valuable suggestions. Following your advice, we have organized this content into a table and relocated it to the Supplementary Text 2. In this study, we used these days to identify the optimal windows for detecting transplanting and harvest dates, specifically, the minimum VH and peak NDYI were extracted from the ranges of these days. This implementation aims to ensure the extraction of the minimum VH and peak NDYI, as any bias in peak or valley could lead to errors in determining transplanting or harvest dates, especially in the case of double and triple rice croppings. The revised content in the manuscript is as follows:

➢ To identify the optimal window for detection of the transplanting and harvest dates, the time window for detection of the minimum VH and peak NDYI were used (Table S1). ~~To identify the optimal time window for detection of the transplanting and harvest dates, the time window for detection of the minimum VH was set from 120 days before the date of peak EVI ($DOY_{EVI_{max}}$) to 45 days before the date of peak EVI, i.e., $\left[DOY_{EVI_{max}} - 120, DOY_{EVI_{max}} - 45\right]$ or from the first day of EVI arc ($DOY_{EVI\,arc_{first\,day}}$) to 45 days before the date of peak EVI, i.e., $\left[DOY_{EVI\,arc_{first\,day}}, DOY_{EVI_{max}} - 45\right]$. The time window for detection of the peak NDYI was set from 13 days after the peak EVI date to 55 days after the date of~~

peak EVI, i.e., $\left[DOY_{EVI_{max}} + 13, DOY_{EVI_{max}} + 55\right]$

$\left[DOY_{EVI_{max}} + 13, DOY_{EVI\,arc_{last\,day}}\right]$ (Lines 225-232).

The revised content in the Supplementary Text 2 is as follows:

➢ **2. Optimal time window for transplanting and harvest dates detection**

To identify the optimal time window for detection of the transplanting and harvest dates, the time window for detection of the minimum VH was set from 120 days before the date of peak EVI to 45 days before the date of peak EVI ($DOY_{EVI_{max}}$), or from the first day of EVI arc ($DOY_{EVI\,arc_{first\,day}}$) to 45 days before the date of peak EVI. The time window for detection of the peak NDYI was set from 13 days after the peak EVI date to 55 days after the date of peak EVI, or from 13 days after the peak EVI date to the last day of the EVI arc ($DOY_{EVI\,arc_{last\,day}}$). It can be shown as follow table.

**Table S1.** Optimal time window for transplanting and harvest dates detection

| Time window | Transplanting date | Harvest date |
|---|---|---|
| 1 | $\left[DOY_{EVI_{max}} - 120, \quad DOY_{EVI_{max}} - 45\right]$ | $\left[DOY_{EVI_{max}} + 13, \quad DOY_{EVI_{max}} + 55\right]$ |
| 2 | $\left[DOY_{EVI\,arc_{first\,day}}, \quad DOY_{EVI_{max}} - 45\right]$ | $\left[DOY_{EVI_{max}} + 13, \quad DOY_{EVI\,arc_{last\,ay}}\right]$ |

■ *Line 200 – what is EVI arc?*

**Response:** The EVI arc denotes the arc with the shape of downward-opening patterns from the smoothed EVI time series. We have addressed it on the revised content as follows:

➢ The six-parametric Weibull function,  can be used to identify the number of rice croppings by depicting an arc with the shape of downward-opening patterns from the smoothed EVI time series (hereafter referred to as EVI arc) (Fig. 2 Step 1 Algorithm b), as shown follows: (Lines 236-239).

■ *Line 207 – that part as the equation format*

**Response:** Yes, that part should be shown as the equation format. We have revised this part as follows:

➢ The six-parametric Weibull function,  can be used to identify the number of rice croppings by depicting an arc with the shape of downward-opening patterns from the smoothed EVI time series (hereafter referred to as EVI arc) (Fig. 2 Step 1 Algorithm b), as shown follows:

$$w(x) = \left(d + \exp\left(-\left(\frac{x}{e}\right)^{f}\right)\right) \times \left(1 - a \exp\left(-\left(\frac{x}{b}\right)^{c}\right)\right) \tag{3}$$

where $a$, $b$, $c$, $d$, $e$, and $f$ are the free parameters to be fitted (Rolinski et al., 2007) (Lines 236-241).

As the equations of coefficient of determination ($R^2$) (Eq. (3)), bias error (Bias) (Eq. (4)), Mean Absolute Error (MAE) (Eq. (5)), and Root Mean Square Error (RMSE) (Eq.

(6)) have been removed, the six-parametric Weibull function has thus been designated as Eq. (3). Consequently, and the equation numbering has been adjusted from 7 to 16 to 4 to 13. Correspondingly, the numbering of equations in the text has been revised as follows:

➢ After application of the function (Eq. (4)), all available arcs of the smoothed EVI time series were then labelled, including the start (start day of detected EVI arc, $DOY_{EVI\ arc_{first\ day}}$), peak (peak day of detected EVI arc, $DOY_{EVI_{max}}$), and end (end day of detected EVI arc, $DOY_{EVI\ arc_{last\ day}}$) of the arc, and the peak EVI value ($Value_{EVI_{max}}$) (Fig. 2 Step 1 Process b) (Lines 252-254).

➢ where $mu$ is the mean direction of the distribution, and $kappa$ is a non-negative numeric value representing a concentration parameter of the distribution; $mu$ and $kappa$ are correspond to $\mu$ and $\kappa$ in Eq. (5), respectively (Lines 303-304).

➢ The circular data $x$ in Eqs. (5) and (6) denote the phenological date shown in DOY format (Line 306).

➢ The parameters $DOY_{integrated}$ and $Var$ were derived from the mle.vonmises function (Eq. (9)), representing the value and variance of the phenological dates ($DOY_{rad}$), respectively (Lines 319-320).

➢ $Var$ parameter, derived from the mle.vonmises function (Eqs. (9) and (11)), is prone to bias, requiring bias-corrected estimates when the sample size is less than 16 (Best and Fisher, 1981) (Lines 520-522).

■ *Line 210 – From R: "To cite package 'cardidates' in publications use: Rolinski, S., Horn, H., Petzoldt, T., Paul, L. (2007). Identification of cardinal dates in phytoplankton time series to enable the analysis of long-term trends. Oecologia 153, 997--1008. doi:10.1007/s00442-007-0783-2"*

**Response:** Thanks for your suggestion. We have cited the package "cardidates" using the following reference:

Rolinski, S., Horn, H., Petzoldt, T., and Paul, L.: Identifying cardinal dates in phytoplankton time series to enable the analysis of long-term trends. Oecologia 153, 997-1008. https://doi.org/10.1007/s00442-007-0783-2, 2007.

Consequently, the revised contents are as follows:

➢ This fitted Weibull function can be implemented using the peakwindow function in the cardidates" package of R  (Rolinski et al., 2007) (Fig. S3a; Fig. 3) (Lines 242-244).

Correspondingly, the unreferenced sources have been removed from the References section:

➢ .

➢ .

➢ .

■ *Figure 3 description: there is no grey area in the charts*

**Response:** We apology for the error. We have revised the caption of Figure 3 as follows:

➢ **Figure 3.** Smoothed EVI time series and subsequent identification of the number of rice croppings at adjacent grids (32.25°N, 130.25°E, and 32.75°N, 130.25E°) across two years. Left column shows the smoothed EVI time series using the

LOESS method. Black points and green lines indicate the EVI value at specific dates and the smoothed EVI time series, respectively. Green area indicates the 95 % confidence interval around the smoothed EVI time series. Right column displays the number of rice croppings detected using the fitted Weibull function implemented via the "cardidates" package in R. Blue, yellow, red, and black lines correspond to the detected first, second, third, and fourth arcs of the smoothed EVI time series (Lines 261-266).

- *Line 265 – cite the 'circular' package properly*

**Response:** Thanks. We have cited the "circular" package as follows:

➢ The availability of the "circular" R package (https://rdrr.io/rforge/circular/man/circular-package.html) (Agostinelli and Lund, 2023) is convenient for analysis of circular data (Lines 299-300).

This new reference has been added to the "References" section:

➢ Agostinelli, C., and Lund, U.: R package "circular": Circular Statistics (version 0.5−0). https://CRAN.R-project.org/package=circular, 2023 (Lines 611-612).

- *The figures 4-7 – parts of the figures' captions are redundant as they include the information already provided in the figure (e.g., what is in upper, middle lower panels etc.)*

**Response:** We've fully accepted your suggestions and removed the redundant parts in the captions from Figure 4 to Figure 7 as follows:

➢ **Figure 4.** Transplanting date and harvest date for the three groups.  Colour gradient from blue to red in the legend denotes the respective transplanting and harvest dates (Lines 347-350).

➢ **Figure 5.** Variance in transplanting date and harvest date for the three groups.  Colour gradient from blue to red in the legend denotes the respective variance in transplanting and harvest dates (Lines 352-355).

➢ **Figure 6.** Transplanting date and harvest date for three rice croppings.  Colour gradient from blue to red in the legend denotes the respective transplanting and harvest dates (Lines 366-369).

➢ **Figure 7.** Variance in transplanting date and harvest date for three rice croppings.  Colour gradient from blue to red in the legend denotes the respective variance in transplanting and harvest dates (Lines 372-375).

At the same time, we have removed the redundant parts in the captions from Figure S5 to Figure S7 in the Supplementary material as follows:

➢ **Figure S6.** Transplanting and harvest dates from the RiceAtlas calendar.  Colour gradient from blue to red in the legend denotes the respective transplanting and harvest dates.

➢ **Figure S6.** Transplanting and harvest dates from the RICA calendar.  Colour gradient from blue to red in the legend denotes the respective transplanting and harvest dates.

➢ **Figure S8.** Transplanting and harvest dates from the SAGE calendar.  Colour gradient from blue to red in the legend denotes the respective transplanting and harvest dates.

■ *Lines 303-305 – maybe add median dates to the figures as well*

**Response:** We sincerely appreciate the reviewer's comments. The median date for each cropping here was statistically calculated from all grids across monsoon Asia. In other words, it was derived from the results presented in Fig. 4 and Fig. 6. The phenological date in each grid of Fig. 4 is obtained from the mean of the 18 grids (3 × 3 grids × 2

years) using the von Mises maximum likelihood estimates methods shown in Step 2. So we keep it as it.

■ *Figure 9 caption – there is no grey dotted line*

**Response:** We apologize for the error. The "Grey dotted line" should be replaced with the "Black solid line". The revised caption of Figure 9 is as follows:

➢ **Figure 9.** Comparison of transplanting date and harvest date for single rice cropping between the proposed rice calendar and the RiceAtlas rice calendar. Blue and orange represent the transplanting date and harvest date, respectively; vertical lines denote the range of the transplanting and harvest dates for the proposed rice calendar; horizontal lines denote the range of the transplanting and harvest dates of the RiceAtlas rice calendar; dots denote the peak of the transplanting or harvest dates. Black dots denote the detected phenological day that falls within the transplanting or harvest ranges from the RiceAtlas rice calendar. Red and black solid lines represent the 1:1 line and regression, respectively (Lines 396-401).

---

## Author Response (AR2)

26 March 2024

**Re: Ms. Ref. No.: essd-2023-283**

Dear Editor and Reviewers,

We would like to thank the editor for handling our manuscript and the reviewers for their valuable comments and constructive suggestions, which have further improved the quality of this manuscript. We have the pleasure of enclosing a revised version of the manuscript "Monsoon Asia Rice Calendar (MARC): a gridded rice calendar in monsoon Asia based on Sentinel-1 and Sentinel-2 images" (Manuscript number: essd-2023-283) and a detailed response to the Reviewers' comments below. We hope that the revised manuscript has been strengthened, addressing the concerns raised.

In the responses below, we have addressed each of the Reviewers' comments in detail. The comments from each Reviewer are noted as "R" (e.g., R2) while each comment is noted as "C" (e.g., C1) to better index all comments. The line numbers indicated refer to the revised manuscript (with track changes). All the changes are highlighted in red with grey background in revised manuscript.

Please do not hesitate to contact us if you require any further information.

Sincerely yours,
Xin Zhao and Kazuya Nishina

**Response to the anonymous reviewer's comments**

*Report #1:*

*Anonymous referee #2:*

*R2C1: Thank you for adressing my comments. The manuscript has been significantly improved and the methods section is much more clear. I have few minor comments:*

We greatly appreciate your valuable and constructive comments, which contributes improving the overall quality of our manuscript. Please see below the point-by-point response, with *your Comments in italic black*, our Responses in blue and Changes to the manuscript in red with grey background.

*Lines 22-24 maybe you could remove 'rice' which is used 5 times in this sentence – for example, "the proposed calendar" or "our calendar" etc.*

**Response:** Thank you for your suggestion. We have removed the "rice" in this sentence. The revised contents are as follows:

➢ When compared with single rice data from the census-based RiceAtlas  calendar, the proposed  calendar  exhibited better results than the MODIS-based RICA  calendar (Lines 22-24).

*R2C2: Do not use space between number and "%"*

**Response:** Thank you for your reminder. We have removed the space between the number and % in the revised manuscript as follows:

- Specifically, concern regarding the negative impacts of rice cultivation is increasing because irrigated rice paddy field is an important source of anthropogenic GHG emissions, contributing 8% and 11% of global methane and nitrous oxide emissions, respectively (Saunois et al., 2020; Jiang et al., 2019) (Lines 37-39).

- To accurately estimate GHG emissions related to rice cultivation and to establish appropriate reduction measures, a detailed rice calendar that depicts rice phenology dynamics is urgently needed, especially for monsoon Asia, which accounts for 87% of the area of harvested rice globally and for 90% of global rice production (FAOSTAT, 2022) (Lines 40-43).

- Invalid observations of Sentinel-2 images caused by clouds and cirrus were removed using cloud filtering (>50%) and the cloud-score method (QA60 quality assessment band with 60 m resolution) (Inoue et al., 2020) (Lines 134-136).

- Flooding rice cultivation, common in Asia and accounting for over 12% of the global cropland (FAOSTAT, 2020; Zhang et al., 2021a), presents a distinctive flooding signal that can be used for detection of rice transplanting date (Lines 168-169).

- Green area indicates the 95% confidence interval around the smoothed EVI time series (Lines 212-213).

- The proposed rice calendar extracts 9% of triple rice croppings (Fig. 11a), which are scattered and distributed in South China, Southeast Asia, and India (Fig. 10a) (Lines 386-387).

- This proportion is close to that of the RICA rice calendar (6% in Fig. 11c), but markedly lower than that of the RiceAtlas rice calendar (41% in Fig. 11c) (Lines 387-388).

*R2C3: Lines 46-48 – something is missing in this sentence*

Response: Thank you for pointing out the mistake. We have rephased this sentence as follows:

➢ The limited number of global rice calendars (e.g., SAGE (Sacks et al., 2010), MIRCA2000 (Portmann et al., 2010), and RiceAtlas (Laborte et al., 2017)) that are currently available, which rely relies on compilation of statistical data at national and/or sub-national scales (Lines 46-48).

*R2C4: Line 84-85 there are many different methods used besides thresholds, such as derivatives or inflection points*

Response: We greatly appreciate the reviewer's constructive suggestions. As you suggested, we have included the other methods (e.g., derivatives and inflection points) into following revised sentence:

➢ Different from most widely used peak greenness detection methods, which depend on thresholds, derivatives, or inflection points for detection (Xin et al., 2020; Yang et al., 2020), the fitted Weibull function omits the noisy peaks, which means it can track the shape of the vegetation index time series (Lines 84-86).

*R2C5: Line 101 - "which covered" -> "which covers". What is the total area?*

Response: We greatly appreciate the reviewer's suggestion. As you suggested, "which covered" has been revised to "which covers" as follows:

➢ The analysed area is located in monsoon Asia, which covers the region of 10° S to 53.5° N, 61° E to 153° E (Line 102).

The total area of monsoon Asia is 2106 millions of ha, which has been added into the revised manuscript as follows:

➢ The total area of monsoon Asia is 2106 millions of ha (Lines 102-103).

*R2C6: Lines 181-182 – please rephrase*

**Response:** Our apologies for the ambiguous description of the sentence. We have rephrased it as follows:

➢  If the peak NDYI could not be obtained from those time windows, peak NDYI was identified using the peak EVI date ($DOY_{EVI_{max}}$) plus the difference days. The difference days for each rice cropping can be found in Zhao et al., (2023) (Fig.2 Step 1 Process a) (Lines 182-186).

*R2C7: Line 196 - "then" is redundant*

**Response:** Thanks for your suggestion again. We have removed "then" as follows:

➢ After application of the function (Eq. (4)), all available arcs of the smoothed EVI time series were  labelled, including the start (start day of detected EVI arc, $DOY_{EVI\ arc_{first\ day}}$), peak (peak day of detected EVI arc, $DOY_{EVI_{max}}$), and end (end day of detected EVI arc, $DOY_{EVI\ arc_{last\ day}}$) of the arc, and the peak EVI value ($Value_{EVI_{max}}$) (Fig. 2 Step 1 Process b) (Lines 201-203).

*R2C8: You use R2 in the figure 9, but is not mentioned in the text. Could you also add that?*

**Response:** We greatly appreciate the reviewer's comments and have added the $R^2$ value in the revised manuscript:

➢ The transplanting dates of the proposed rice calendar are consistent with those of the RiceAtlas rice calendar, with $R^2$ of 0.43, Bias of 3.93 days, MAE of 16.38 days, and RMSE of 27.62 days (Fig. 9). Additionally, the harvest dates of the proposed rice calendar are correlated with those of the RiceAtlas rice calendar, with $R^2$ of 0.44, Bias of -5.76 days, MAE of 17.87 days, and RMSE of 28.32 days (Fig. 9) (Lines 324-327).

*Report #2:*

*Anonymous referee #3:*

*This study presents a new gridded rice calendar for monsoon Asia spanning from 2019 to 2020, with a resolution of 0.5° × 0.5°, utilizing Sentinel-1 and Sentinel-2 satellite imagery. The novelty of this rice calendar lies in its development of a consistent and optimal methodological framework, enabling the spatially explicit characterization of rice transplanting dates, harvest dates, and the number of rice croppings. This framework comprises two key steps: the detection of rice phenological dates and the number of rice croppings using a feature-based algorithm and the fitted Weibull function, followed by the spatio-temporal integration of the detected dates using von Mises maximum likelihood estimates. The development of the gridded rice calendar for monsoon Asia represents an advancement in agricultural research, offering a valuable resource for researchers and policymakers alike. Generally the work is well done. However, I have some comments before its consideration for publication.*

We greatly appreciate your valuable and constructive comments, which contributes improving the overall quality of our manuscript. Please see below the point-by-point response, with *your Comments in italic black*, our Responses in blue and Changes to the manuscript in red with grey background.

*Major comments:*

*R3C1: The authors' response to the reviewer's comment regarding the spatial resolution of the proposed rice calendar is not convinced. Why not producing the 10-m phenology data? It would be valuable if the original 10-m phenology information from Sentinel-1 and -2 data can be released and shared in this study.*

**Response:** We appreciate the reviewer's important comments. We believe that increasing spatial resolution is a future challenge, but in this case we have created this information for use

in global biogeochemical and crop models. Regarding the decision to produce a 0.5° resolution calendar instead of a 10-m resolution calendar, we based our decision on the following considerations, as explained in detail below:

(1) There is research gap in the spatial resolution of rice calendars for large areas. Given the currently available global/continental rice calendars, such as RiceAtlas, RICA, and SAGE, which are all based on national/subnational scales, we aimed to produce a large-scale rice calendar with a finer resolution than the existing rice calendars. In other words, the spatial resolution of our proposed rice calendar surpasses that of existing large-scale rice calendars, which can be considered as an advantage of our proposed rice calendar.

(2) The production of 0.5° resolution rice calendar fulfills the requirements for land surface model/terrestrial process-based model simulations. One of the potential and important application of our proposed rice calendar is as an input to land surface model or terrestrial process-based model for estimating greenhouse gas emissions or rice production. Such models are typically simulated at 0.5° resolution, with examples like LPJ-GUESS (Smith et al., 2014), VISIT (Ito, 2019), DLEM (Tian et al., 2009), ORCHIDEE-CROP (Müller et al., 2019), and ISAM (Lin et al., 2021).

(3) The production of 0.5° resolution rice calendar is the result of a trade-off between depicting rice phenology at large scale and computational sources constraints. We agree that 10-m resolution rice calendar would be valuable, however, there are some practical limitations at global (Asian) scale. Processing the Sentinel-1&-2 images at 10-m resolution for monsoon Asia would require immense computational power. In this study, we had to process $127 \times 184 = 23,368$ grids $\times$ 2 years $= 46,736$ grids at each process, including extracting time series of VH, EVI, and NDYI from Sentinel-1&-2 images, smoothing time series data, and identifying the phenological dates. If we were to prefer 10-m resolution, we would have to process $705,842 \times 888,631 = 6.275 \times 10^{11}$ grids $\times$ 2 years $= 12.55 \times 10^{11}$ grids, which accounts for approximately $2.683 \times 10^7$ times more than current study. Also, in practice, the satellite imagery we use faces issues due to cloud coverage, and it is difficult for the algorithm to work well on all grids at detailed scales. Therefore, to produce rice calendars

with detailed spatial resolution, high-frequency data sources such as constellations or geostationary satellites may be required, which is outside the scope of this paper.

Additionally, the primary objective of our study was to develop a continental-scale rice calendar that could provide a synoptic view of rice phenology across monsoon Asia. While 10-m resolution rice phenology data would be beneficial for local-scale applications, it might not be essential for capturing the overall regional patterns and variability in rice phenology. Instead, we had provided the variance in transplanting and harvest dates for each grid, as shown in Fig. 7. Most of phenological dates in each grid vary by less than 50 days, or even 20 days, which could indicate a small variability of phenological dates within each grid.

Optimistically, our proposed rice calendar provides a feasible methodological framework, which enables future researchers to utilize this methodology for high-resolution rice calendar production while minimizing the computational requirements.

References:

Smith, B., Warlind, D., Arneth, A., Hickler, T., Leadley, P., Siltberg, J., Zaehle, S.: Implications of incorporating N cycling and N limitations on primary production in an individual-based dynamic vegetation model, Biogeosciences, 11, 2027–2054, https://doi.org/10.5194/bg-11-2027-2014, 2009.

Ito, A.: Disequilibrium of terrestrial ecosystem $CO_2$ budget caused by disturbance-induced emissions and non-$CO_2$ carbon export flows: a global model assessment, Earth Syst. Dynam., 10, 658–709, https://doi.org/10.5194/esd-10-685-2019, 2019.

Tian, H., Chen, G., Liu, M., Zhang, C., Sun, G., Lu, C., Xu, X., Ren, W., Pan, S., Chappelka, A.: Model estimates of net primary productivity, evapotranspiration, and water use efficiency in the terrestrial ecosystems of the southern United States during 1895-2007, For. Ecol. Manag., 259, 1311–1327, https://doi.org/10.1016/j.foreco.2009.10.009, 2009.

Müller, C., Elliott, J., Kelly, D., Arneth, A., Balkovic, J., Ciais, P., Deryng, D., Folberth, C., Hoek, S., Izaurralde, R.C., Jones, C.D., Khabarov, N., Lawrence, P., Liu, W., Olin, S., Pugh, T.A.M., Reddy, A., Rosenzweig, C., Ruane, A.C., Sakurai, G., Schmid, E., Skalsky, R., Wang, X., Wit, A., Yang, H.: The global gridded crop model intercomparison phase 1 simulation dataset, Sci. Data, 6, 50, https://doi.org/10.1038/s41597-019-0023-8, 2019.

Lin, T., Song, Y., Lawrence, P., Kheshgi, H.S., Jain, A.K.: Worldwide maize and soybean yield response to environmental and management factors over the 20th and 21st centuries, J. Geophys. Res. Biogeosci., 6, 50, https://doi.org/10.1029/2021JG006304, 2021.

*R3C2: The validation of the produced phenology data is still not durable. The Census-based RiceAtlas rice calendar actually cannot be used as "ground truth" data. Can the authors collect some in-situ phenology data for validation? For example, some PhenoCAM-based phenology data can be used for validation of at least harvest timing.*

**Response:** We sincerely appreciate and agree with your comments. As you pointed out, the census-based RiceAtlas rice calendar is not "ground truth" data, which was also emphasized in our previous paper (Zhao et al., 2023).

In response to this concern, we have taken series processes to conduct the site validation. Firstly, we have collected 39 in-situ records of rice transplanting and harvest dates from the literatures. The years of these selected records are very close to the years used for producing our proposed rice calendar. Additionally, as suggested, we have obtained one rice paddy site in monsoon Asia from the PhenoCam dataset, covering the same experimental years as our study. The available wavelengths from PhenoCam dataset were used to calculate the NDYI time series and detect the harvest date. In total, we have collected 40 records for site validation, covering most areas of monsoon Asia, which can be considered representative. The geographic locations, transplanting dates, and harvest dates of these records are summarized in Table S2 of the Supplementary Text 3. The distribution of these records is shown in Fig. S11. We emphasized

in the manuscript, however, that even these site observations cannot be considered ground truth, as the sites may not be representative of the entire area (often being cultivation-managed for research) and these site observations were made in different years from the satellite images.

We have added the description of site validation on Supplementary Text 3 as follows:

**➢ 3. Site validation**

To further validate the proposed rice calendar, site phenological dates close to the experimental period were collected from two sources: 1) 39 sites recorded in the literatures, and 2) observations from one site in the PhenoCam dataset. The transplanting and harvest dates were directly extracted from the literature records for the 39 sites. Since there is only one rice paddy site located in monsoon Asia in the PhenoCam dataset, the Jurong site provides a time series of vegetation phenological observations derived from conventional visible-wavelength automated digital camera imagery. The transplanting and harvest dates for all 40 sites are summarized in Table S2. These 40 sites are representative due to their wide coverage across monsoon Asia (Fig. S11).

[Figure]

**Figure S11.** Location of the validation sites in monsoon Asia. Green areas indicate rice paddy fields, and bold black borders indicate the countries in this study area. Yellow circles denote the validation site collected from the literatures and dataset.

**Table S2.** Transplanting date and harvest date for 40 sites, along with the corresponding phenological dates from rice calendars at each site location

| Country | Latitude | Longitude | T_site | H_site | T_MARC | H_MARC | T_RiceAtlas | H_RiceAtlas | T_RICA | H_RICA | T_SAGE | H_SAGE | Reference |
|---------|----------|-----------|--------|--------|--------|--------|-------------|-------------|--------|--------|--------|--------|-----------|
| Thailand | 14.01 °N | 101.22 °E | 182 | 273 | 190.34 | 285.34 | 135 | 306 | 138.73 | 264.66 | 185.5 | 339 | Chidthaisong et al., 2018 |
| South Korea | 36.37 °N | 127.33 °E | 149 | 289 | 164.25 | 266.18 | 148 | 275 | 145.08 | 277.41 | 151 | 274 | Choi et al., 2019 |
| China | 30.97 °N | 121.01 °E | 175 | 297 | 175.61 | 269.95 | 160 | 304 | 262.18 | 19.18 | 121.5 | 245.5 | Fang et al., 2021 |
| Japan | 35.71 °N | 140.34 °E | 158 | 266 | 157.68 | 251.79 | 117 | 244 | 124.19 | 252.35 | 167 | 291 | Fawibe et al., 2019 |
| China | 32.10 °N | 112.40 °E | 152 | 274 | 159.85 | 253.46 | 166 | 294 | 136.64 | 257.02 | 121.5 | 245.5 | Feng et al., 2021 |
| Bangladesh | 24.75 °N | 90.50 °E | 20 | 119 | 34.10 | 128.45 | 5 | 110 | 17.83 | 129.72 | -12 | 127.5 | Forhad et al., 2019 |
| China | 30.21 °N | 112.09 °E | 157 | 257 | 150.93 | 250.49 | 166 | 294 | 136.64 | 257.02 | 121.4 | 245.5 | Fu et al., 2019 |
| South Korea | 38.20 °N | 127.25 °E | 121 | 246 | 155.95 | 260.51 | 140 | 267 | 128.19 | 265.18 | 151 | 274 | Huang et al., 2018 |
| South Korea | 38.20 °N | 127.25 °E | 129 | 257 | 155.95 | 260.51 | 140 | 267 | 128.19 | 265.18 | 151 | 274 | Hwang et al., 2020 |
| Philippines | 14.16 °N | 121.26 °E | 30 | 133 | -28.93 | 68.77 | −16 | 105 | -8.76 | 105.02 | 130 | 301 | Islam et al., 2020 |
| Bangladesh | 23.60 °N | 90.25 °E | 25 | 120 | 52.86 | 152.85 | 10 | 105 | 173.01 | 285.63 | -12 | 127.5 | Islam et al., 2020 |
| Bangladesh | 24.44 °N | 90.24 °E | 23 | 118 | 37.16 | 134.05 | 15 | 120 | 50.36 | 154.87 | -12 | 127.5 | Islam et al., 2020 |

| Country | Latitude | Longitude | | | | | | | | | | | Reference |
|---|---|---|---|---|---|---|---|---|---|---|---|---|---|
| South Korea | 38.20 ºN | 127.25 ºE | 135 | 257 | 155.95 | 260.51 | 140 | 267 | 128.19 | 265.18 | 151 | 274 | Jeong et al., 2020 |
| South Korea | 34.48 ºN | 126.48 ºE | 152 | 306 | 167.17 | 265.46 | - | - | - | - | 151 | 274 | Jeong et al., 2020 |
| China | 22.88 ºN | 108.29 ºE | 102 | 199 | 86.25 | 172.03 | 101 | 195 | 88.08 | 210.59 | 88 | 179.5 | Li et al., 2020 |
| China | 30.14 ºN | 115.25 ºE | 121 | 229 | 161.46 | 252.94 | 100 | 181 | 136.64 | 257.02 | 121.5 | 245.5 | Liang et al., 2019 |
| China | 28.44 ºN | 116.00 ºE | 195 | 304 | 186.04 | 287.20 | 140 | 260 | 109.38 | 252.92 | 182.5 | 306 | Liu et al., 2019a |
| China | 28.10 ºN | 116.50 ºE | 116 | 203 | 104.35 | 195.59 | 105 | 201 | 109.38 | 252.92 | 88 | 179.5 | Liu et al., 2019b |
| China | 31.22 ºN | 104.62 ºE | 149 | 268 | 175.62 | 269.52 | 135 | 270 | 99.7 | 235.3 | 121.5 | 245.5 | Liu et al., 2021 |
| Thailand | 14.37 ºN | 100.61 ºE | 305 | 57 | 309.77 | 43.72 | 390 | 135 | 338.71 | 103.1 | 37.5 | 148.5 | Maneepitak et al., 2019 |
| Japan | 43.18 ºN | 141.44 ºE | 144 | 258 | 175.09 | 268.64 | - | - | - | - | 167 | 291 | Naser et al., 2020 |
| China | 46.95 ºN | 127.67 ºE | 139 | 264 | 155.51 | 256.52 | 135 | 266 | 137.53 | 261.92 | 121.5 | 245.5 | Nie et al., 2020 |
| Indonesia | -7.79 ºN | 111.10 ºE | 102 | 203 | 69.22 | 160.35 | 130 | 248 | - | - | 151 | 243 | Nugroho et al., 2018 |
| Japan | 36.03 ºN | 140.11 ºE | 140 | 271 | 150.23 | 257.62 | 126 | 251 | 132.97 | 256.28 | 167 | 291 | Okamura et al., 2018 |
| India | 11.00 ºN | 79.50 ºE | 167 | 264 | 194.62 | 310.46 | 181 | 301 | 199.21 | 315.07 | 133.5 | 231.5 | Oo et al., 2020 |
| China | 26.45 ºN | 111.52 ºE | 116 | 199 | 111.23 | 198.67 | 110 | 200 | 109.94 | 240.2 | 88 | 179.5 | Raheem et al., 2019 |
| Indonesia | -6.78 ºN | 111.20 ºE | 92 | 196 | 93.48 | 183.12 | 130 | 248 | - | - | 151 | 243 | Setyanto et al., 2018 |

| Country | Lat | Lon | T_site | H_site | T_MARC | H_MARC | T_RiceAtlas | H_RiceAtlas | T_RICA | H_RICA | T_SAGE | H_SAGE | Reference |
|---|---|---|---|---|---|---|---|---|---|---|---|---|---|
| Philippines | 15.67 °N | 120.90 °E | 168 | 260 | 187.61 | 289.05 | 189 | 285 | 192.15 | 294.22 | 130 | 301.5 | Sibayan et al., 2018 |
| China | 31.16 °N | 119.54 °E | 160 | 313 | 171.34 | 267.08 | 166 | 280 | 75.83 | 186.61 | 121.5 | 245.5 | Sun et al., 2019a |
| China | 39.88 °N | 123.58 °E | 149 | 262 | 159.20 | 264.02 | 140 | 284 | 141.37 | 264.41 | 121.5 | 245.5 | Sun et al., 2019b |
| Vietnam | 16.47 °N | 107.52 °E | 20 | 140 | 52.27 | 143.62 | 30 | 140 | 7.2 | 134.67 | 18 | 113.5 | Tran et al., 2018 |
| Vietnam | 16.47 °N | 107.52 °E | 162 | 252 | 150.62 | 247.52 | 155 | 265 | 147.96 | 242.73 | 227 | 365 | Tran et al., 2018 |
| China | 32.86 °N | 117.40 °E | 180 | 301 | 173.67 | 271.41 | 161 | 274 | 155.72 | 265.26 | 121.5 | 245.5 | Wang et al., 2020 |
| China | 32.21 °N | 118.71 °E | 170 | 299 | 176.84 | 271.19 | 166 | 280 | 75.83 | 186.61 | 121.5 | 245.5 | Wang et al., 2021 |
| China | 43.32 °N | 123.23 °E | 149 | 289 | 155.60 | 253.35 | 105 | 227 | 131.48 | 261.93 | 121.5 | 245.5 | Wu et al., 2019 |
| China | 31.25 °N | 120.96 °E | 181 | 304 | 175.61 | 269.95 | 166 | 280 | 75.83 | 186.61 | 121.5 | 245.5 | Yang et al., 2018 |
| China | 32.58 °N | 119.70 °E | 173 | 307 | 181.15 | 277.72 | 166 | 280 | 75.83 | 186.61 | 121.5 | 245.5 | Yuan et al., 2021 |
| China | 32.50 °N | 119.42 °E | 164 | 292 | 182.86 | 279.63 | 166 | 280 | 75.83 | 186.61 | 121.5 | 245.5 | Zhang et al., 2018 |
| China | 32.30 °N | 119.25 °E | 164 | 294 | 179.19 | 277.27 | 166 | 280 | 75.83 | 186.61 | 121.5 | 245.5 | Zhang et al., 2019 |
| China (Jurong, PhenoCam site) | 119.22 °N | 31.81 °E | - | 273.6 | 177.84 | 274.33 | - | 280 | 75.83 | 186.61 | 121.5 | 245.5 | Seyednasrollah et al., 2019 |

Note: T_site and H_site denote the transplanting date and harvest date of sites from literatures and dataset. T_MARC and H_MARC denote the transplanting date and harvest date of the proposed rice calendar at each site location. T_RiceAtlas and H_RiceAtlas denote the transplanting date and harvest date of the RiceAtlas rice calendar at each site location. T_RICA and H_RICA denote the transplanting date and harvest date of the RICA rice calendar at each site location. T_SAGE and H_SAGE denote the transplanting date and harvest date of the SAGE rice calendar at each site location. '-' denotes phenological dates are not available. Phenological dates less than 0 indicate that the day has been subtracted by 365 days for easy comparison.

The hyperspectral image data has been analyzed in detail in our previous paper (Zhao et al., 2023). Considering the concerns about the length of this manuscript from Reviewer#2, the PhenoCam data processing is described as follows:

> **Processing of PhenoCam data**

Among the 393 sites across diverse ecosystems worldwide in the PhenoCam dataset, the Jurong Observation Station (JROS) is the only rice paddy field site located in monsoon Asia (Seyednasrollah et al., 2019). All available images from PhenoCam from 1 January 2019 to 31 December 2020 were used for detecting harvest dates. To calculate the NDYI vegetation index, green and blue bands were used based on Eq. (2) in the manuscript. The Locally Estimated Scatterplot Smoothing (LOESS) method was adopted to smooth the NDYI time series (Fig. S12). The span value was assigned as 0.2 to depict the NDYI time series pattern. The peak NDYI was detected, and the day on which the peak NDYI occurred was identified as the harvest date. Thus, DOY 273.44 was detected as showing the peak NDYI and identified as the harvest date at Jurong site (Table S2).

[Figure]

**Figure.** Smoothed NDYI time series and identification of peak NDYI at Jurong site from PhenoCam dataset. Black points and orange lines indicate the NDYI value at specific dates and the smoothed NDYI

time series, respectively. Orange area indicates the 95% confidence interval around the smoothed NDYI time series. Red vertical line indicates the peaks of NDYI.

At the same time, we have extracted the transplanting and harvest dates from the proposed rice calendar based on the locations of these 40 records, as shown in Table S2.

The proposed rice calendar demonstrates high agreement with the site phenological dates (Fig. S12), with $R^2$ of 0.90 and 0.87, Bias of 7.99 and −9.07 days, MAE of 16.32 and 19.58 days, and RMSE of 19.00 and 22.43 days for transplanting date and harvest date, respectively. This site validation underscores the robustness of proposed rice calendar. The results of site validation have been added in the Supplementary Text 3.1 as follows:

➢ **3.1 Comparison of transplanting and harvest dates between proposed rice calendar and site records**

The transplanting and harvest dates from the proposed rice calendar were further compared with those from the site records. The transplanting and harvest dates were firstly extracted from the proposed rice calendar at each site location as shown in Table S2. The transplanting dates of the proposed rice calendar are consistent with those site records, with $R^2$ of 0.9, Bias of 7.99 days, MAE of 16.32 days, and RMSE of 19.00 days (Fig. S12). Additionally, the harvest dates of the proposed rice calendar are correlated with those of the site records with $R^2$ of 0.87, Bias of −9.07 days, MAE of 19.58 days, and RMSE of 22.43 days (Fig. S12). This site validation demonstrates the efficacy of the proposed rice calendar.

[Figure]

**Figure S12.** Comparison of transplanting date and harvest date between the proposed rice calendar and site records. Blue and orange points represent the transplanting date and harvest date, respectively. Red and black solid lines represent the 1:1 line and regression, respectively.

Furthermore, to further emphasize the robustness of proposed rice calendar and to evaluate the advantage of the proposed rice calendar, we have compared the site validation results with those of other existing rice calendars, including RiceAtlas, RICA, and SAGE. We have extracted the transplanting and harvest dates from these three rice calendars based on the same locations as the 40 validation records (Table S2). The results, shown in Figure S13, demonstrate that our proposed rice calendar performs better than the other rice calendars, with higher agreement with the in-situ phenological dates in terms of $R^2$, Bias, MAE, and RMSE (except for the Bias of the RiceAtlas rice calendar). This site validation comparison highlights the advantage and progress of our proposed rice calendar.

➢ **3.2 Comparison of transplanting and harvest dates between other rice calendars and site records**

To evaluate the advantage of the proposed rice calendar, the transplanting and harvest dates from other rice calendars were compared with those from the site records. The transplanting and harvest dates from RiceAtlas, RICA, and SAGE rice calendars were extracted at each site location, as shown in Table S2. The transplanting dates of RiceAtlas, RICA, and SAGE rice calendars were correlated with the site records, with $R^2$ of 0.86, Bias of −2.41 days, MAE of 19.10 days, and RMSE of 26.56 days; $R^2$ of 0.42, Bias of −20.85 days, MAE of 41.41 days, and RMSE of 57.66 days; $R^2$ of 0.64, Bias of −11.28 days, MAE of 37.08 days, and RMSE of 45.31 days, respectively (Fig. S13). Similarly, the harvest dates of the RiceAtlas, RICA, and SAGE rice calendars were correlated with site records, with $R^2$ of 0.80, Bias of 2.96 days, MAE of 22.75 days, and RMSE of 29.51 days; $R^2$ of 0.11, Bias of −29.29 days, MAE of 56.25 days, and RMSE of 85.61 days; $R^2$ of 0.44, Bias of −6.88 days, MAE of 43.60 days, and RMSE of 61.10 days, respectively (Fig. S13). The phenological dates of the proposed rice calendar were found to be closer to the site records compared to those of three rice calendars. The good performance in the site validation clearly demonstrates the ability and advantage of the proposed rice calendar in retrieving rice transplanting and harvest dates.

[Figure]

**Figure S13.** Comparison of transplanting date and harvest date between the rice calendars (RiceAtlas, RICA, and SAGE) and site records. Blue and orange points represent the transplanting date and harvest date, respectively. Red and black solid lines represent the 1:1 line and regression, respectively.

**References**

Chidthaisong, A., Cha-un, N., Rossopa, B., Buddaboon, C., Kunuthai, C., Sriphirom, P., Towprayoon, S., Tokida, T., Padre, A. T., and Minamikawa, K.: Evaluating the effects of alternate wetting and drying (AWD) on methane and nitrous oxide emissions from a paddy field in Thailand, Soil Sc. Plant Nutr., 64, 31–38, https://doi.org/10.1080/00380768.2017.1399044, 2018.

Choi, E. J., Jeong, H. C., Kim, G. Y., Lee, S. I., Gwon, H. S., Lee, J. S., and Oh, T. K.: Assessment of methane emission with application of rice straw in a paddy field. Korean J. Agric. Sci., 46(4), 857–868. https://doi.org/10.7744/KJOAS.20190069, 2019.

Fang, K., Gao, H., Sha, Z., Dai, W., Yi, X., Chen, H., and Cao, L.: Mitigating global warming potential with increase net ecosystem economic budget by integrated rice-frog farming in eastern China, Agric. Ecosyst. Environ., 308, 107235, https://doi.org/10.1016/j.agee.2020.107235, 2021.

Fawibe, O. O., Honda, K., Taguchi, Y., Park, S., and Isoda, A.: Greenhouse gas emissions from rice field cultivation with drip irrigation and plastic film mulch, Nutr. Cycl. Agroecosyst., 113, 51–62, https://doi.org/10.1007/s10705-018-9961-3, 2019.

Feng, Z. Y., Qin, T., Du, X. Z., Sheng, F., and Li, C. F.: Effects of irrigation regime and rice variety on greenhouse gas emissions and grain yields from paddy fields in central China, Agric. Water Manage., 250, 106830, https://doi.org/10.1016/j.agwat.2021.106830, 2021.

Forhad, A. B. M., Ahatun, R., and Islam, M. Z.: Soil microbial status and methane emission under waste materials application in rice paddy field. International Journal of Environmental Sciences & Natural Resources, 18(1), 555977, https://doi.org/10.19080/IJESNR.2019.18.555977, 2019.

Fu, J., Wu, Y., Wang, Q., Hu, K., Wang, S., Zhou, M., Hayashi, K., Wang, H., Zhan, X., Jian, Y., Cai, C., Song, M., Liu, K., Wang, Y., Zhou, F., and Zhu, J.: Importance of subsurface fluxes of water, nitrogen and phosphorus from rice paddy fields relative to surface runoff, Agric. Water Manage., 213, 627–635, https://doi.org/10.1016/j.agwat.2018.11.005, 2019.

Huang, Y., Ryu, Y., Jiang, C., Kimm, H., Kim, S., Kang, M., and Shim, K.: BESS-Rice: A remote sensing derived and biophysical process-based rice productivity simulation model, Agric. For. Meteorol., 256–257, 253–269, https://doi.org/10.1016/j.agrformet.2018.03.014, 2018.

Hwang, Y., Ryu, Y., Huang, Y., Kim, J., Iwata, H., and Kang, M.: Comprehensive assessments of carbon dynamics in an intermittently-irrigated rice paddy, Agricu. For. Meteorol., 285–286, 107933, https://doi.org/10.1016/j.agrformet.2020.107933, 2020.

Islam, S. F.-u., de Neergaard, A., Sander, B. O., Jensen, L. S., Wassmann, R., and van Groenigen, J. W.: Reducing greenhouse gas emissions and grain arsenic and lead levels without compromising yield in organically produced rice, Agricu. Ecosyst. Environ., 295, 106922, https://doi.org/10.1016/j.agee.2020.106922, 2020.

Jeong, S., Ko, J., Kang, M., Yeom, J., Ng, C. T., Lee, S.H., Lee, Y. G., and Kim, H. Y.: Geographical variations in gross primary production and evapotranspiration of paddy rice in the Korean Peninsula, Sci. Total Environ., 714, 136632, https://doi.org/10.1016/j.scitotenv.2020.136632, 2020.

Li, L., Li, F., and Dong, Y.: Greenhouse Gas Emissions and Global Warming Potential in Double-Cropping Rice Fields as Influenced by Two Water-Saving Irrigation Modes in South China, J. Soil Sci. Plant Nutr., 20, 2617–2630, https://doi.org/10.1007/s42729-020-00328-5, 2020.

Ling, X., Zhang, T., Deng, N., Yuan, S., Yuan, G., He, W., Cui, K., Nie, L., Peng, S., Li, T., and Huang, J.: Modelling rice growth and grain yield in rice ratooning production system, Field Crop. Res., 241, 107574, https://doi.org/10.1016/j.fcr.2019.107574, 2019.

Liu, B., Cui, Y., Luo, Y., Shi, Y., Liu, M., and Liu, F.: Energy partitioning and evapotranspiration over a rotated paddy field in Southern China, Agric. For. Meteorol., 276–277, 107626, https://doi.org/10.1016/j.agrformet.2019.107626, 2019a.

Liu, J., Huang, X., Jiang, H., and Chen, H.: Sustaining yield and mitigating methane emissions from rice production with plastic film mulching technique, Agric. Water Manage., 245, 106667, https://doi.org/10.1016/j.agwat.2020.106667, 2021.

Liu, Y., Tang, H., Muhammad, A., Zhong, C., Li, P., Zhang, P., Yang, B., and Huang, G.: Rice Yield and Greenhouse Gas Emissions Affected by Chinese Milk Vetch and Rice Straw Retention with Reduced Nitrogen Fertilization, Agrono. J., 111, 3028–3038, https://doi.org/10.2134/agronj2019.03.0145, 2019b.

Maneepitak, S., Ullah, H., Datta, A., Shrestha, R. P., Shrestha, S., and Kachenchart, B.: Effects of water and rice straw management practices on water savings and greenhouse gas emissions from a double-rice paddy field in the Central Plain of Thailand, Eur. J. Agron., 107, 18–29, https://doi.org/10.1016/j.eja.2019.04.002, 2019.

Naser, H. M., Nagata, O., Sultana, S., and Hatano, R.: Carbon Sequestration and Contribution of $CO_2$, $CH_4$ and $N_2O$ Fluxes to Global Warming Potential from Paddy-Fallow Fields on Mineral Soil Beneath Peat in Central Hokkaido, Japan, Agriculture, https://doi.org/10.3390/agriculture10010006, 2020.

Nie, T., Chen, P., Zhang, Z., Qi, Z., Zhao, J., Jiang, L., and Lin, Y.: Effects of irrigation method and rice straw incorporation on $CH_4$ emissions of paddy fields in Northeast China, Paddy Water Environ., 18, 111-120, https://doi.org/10.1007/s10333-019-00768-5, 2020.

Nugroho, B. D. A., Toriyama, K., Kobayashi, K., Arif, C., Yokoyama, S., and Mizoguchi, M.: Effect of intermittent irrigation following the system of rice intensification (SRI) on rice yield in a farmer's paddy fields in Indonesia, Paddy Water Environ., 16, 715–723, https://doi.org/10.1007/s10333-018-0663-x, 2018.

Okamura, M., Arai-Sanoh, Y., Yoshida, H., Mukouyama, T., Adachi, S., Yabe, S., Nakagawa, H., Tsutsumi, K., Taniguchi, Y., Kobayashi, N., and Kondo, M.: Characterization of high-yielding rice cultivars with different grain-filling properties to clarify limiting factors for improving grain yield, Field Crop. Res., 219, 139–147, https://doi.org/10.1016/j.fcr.2018.01.035, 2018.

Oo, A. Z., Sudo, S., Fumoto, T., Inubushi, K., Ono, K., Yamamoto, A., Bellingrath-Kimura, S. D., Win, K. T., Umamageswari, C., Bama, K. S., Raju, M., Vanitha, K., Elayakumar, P., Ravi, V., and Ambethgar, V.: Field Validation of the DNDC-Rice Model for Methane and Nitrous Oxide Emissions from Double-Cropping Paddy Rice under Different Irrigation Practices in Tamil Nadu, India, Agriculture, https://doi.org/10.3390/agriculture10080355, 2020.

Raheem, A., Zhang, J., Huang, J., Jiang, Y., Siddik, M. A., Deng, A., Gao, J., and Zhang, W.: Greenhouse gas emissions from a rice-rice-green manure cropping system in South China, Geoderma, 353, 331–339, https://doi.org/10.1016/j.geoderma.2019.07.007, 2019.

Setyanto, P., Pramono, A., Adriany, T. A., Susilawati, H. L., Tokida, T., Padre, A. T., and Minamikawa, K.: Alternate wetting and drying reduces methane emission from a rice paddy in Central Java, Indonesia without yield loss, Soil Sci. Plant Nutr., 64, 23–30, https://doi.org/10.1080/00380768.2017.1409600, 2018.

Seyednasrollah, B., Young, A. M., Hufkens, K., Milliman, T., Friedl, M. A., Frolking, S., Richardson, A. D., Abraha, M., Allen, D. W., Apple, M., Arain, M. A., Baker, J., Baker, J. M., Baldocchi, D., Bernacchi, C. J., Bhattacharjee, J., Blanken, P., Bosch, D. D., Boughton, R., Boughton, E. H., Brown, R. F., Browning, D. M., Brunsell, N., Burns, S. P., Cavagna, M., Chu, H., Clark, P. E., Conrad, B. J., Cremonese, E., Debinski, D., Desai, A. R., Diaz-Delgado, R., Duchesne, L., Dunn, A. L., Eissenstat, D. M., El-

Madany, T., Ellum, D. S. S., Ernest, S. M., Esposito, A., Fenstermaker, L., Flanagan, L. B., Forsythe, B., Gallagher, J., Gianelle, D., Griffis, T., Groffman, P., Gu, L., Guillemot, J., Halpin, M., Hanson, P. J., Hemming, D., Hove, A. A., Humphreys, E. R., Jaimes-Hernandez, A., Jaradat, A. A., Johnson, J., Keel, E., Kelly, V. R., Kirchner, J. W., Kirchner, P. B., Knapp, M., Krassovski, M., Langvall, O., Lanthier, G., Maire, G. l., Magliulo, E., Martin, T. A., McNeil, B., Meyer, G. A., Migliavacca, M., Mohanty, B. P., Moore, C. E., Mudd, R., Munger, J. W., Murrell, Z. E., Nesic, Z., Neufeld, H. S., O'Halloran, T. L., Oechel, W., Oishi, A. C., Oswald, W. W., Perkins, T. D., Reba, M. L., Rundquist, B., Runkle, B. R., Russell, E. S., Sadler, E. J., Saha, A., Saliendra, N. Z., Schmalbeck, L., Schwartz, M. D., Scott, R. L., Smith, E. M., Sonnentag, O., Stoy, P., Strachan, S., Suvocarev, K., Thom, J. E., Thomas, R. Q., Van den berg, A. K., Vargas, R., Verfaillie, J., Vogel, C. S., Walker, J. J., Webb, N., Wetzel, P., Weyers, S., Whipple, A. V., Whitham, T. G., Wohlfahrt, G., Wood, J. D., Wolf, S., Yang, J., Yang, X., Yenni, G., Zhang, Y., Zhang, Q., and Zona, D.: PhenoCam Dataset v2.0: Vegetation Phenology from Digital Camera Imagery, 2000–2018 [data set], ORNL Ridge, https://doi.org/10.3334/ORNLDAAC/1674, 2019.

Sibayan, E. B., Samoy-Pascual, K., Grospe, F. S., Casil, M. E. D., Tokida, T., Padre, A. T., and Minamikawa, K.: Effects of alternate wetting and drying technique on greenhouse gas emissions from irrigated rice paddy in Central Luzon, Philippines, Soil Sci. Plant Nutr., 64, 39–46, https://doi.org/10.1080/00380768.2017.1401906, 2018.

Sun, H., Lu, H., and Feng, Y.: Greenhouse gas emissions vary in response to different biochar amendments: an assessment based on two consecutive rice growth cycles, Environ. Sci. Pollut. Res., 26, 749–758, https://doi.org/10.1007/s11356-018-3636-0, 2019a.

Sun, Y., Xia, G., He, Z., Wu, Q., Zheng, J., Li, Y., Wang, Y., Chen, T., and Chi, D.: Zeolite amendment coupled with alternate wetting and drying to reduce nitrogen loss and enhance rice production, Field Crop. Res., 235, 95–103, https://doi.org/10.1016/j.fcr.2019.03.004, 2019b.

Tran, D. H., Hoang, T. N., Tokida, T., Tirol-Padre, A., and Minamikawa, K.: Impacts of alternate wetting and drying on greenhouse gas emission from paddy field in Central Vietnam, Soil Sci. Plant Nutr., 64, 14–22, https://doi.org/10.1080/00380768.2017.1409601, 2018.

Wang, H., Zhang, Y., Zhang, Y., McDaniel, M. D., Sun, L., Su, W., Fan, X., Liu, S., and Xiao, X.: Water-saving irrigation is a 'win-win' management strategy in rice paddies – With both reduced greenhouse gas emissions and enhanced water use efficiency, Agric. Water Manage., 228, 105889, https://doi.org/10.1016/j.agwat.2019.105889, 2020.

Wang, Y., Hu, Z., Shen, L., Liu, C., Islam, A. R. M. T., Wu, Z., Dang, H., and Chen, S.: The process of methanogenesis in paddy fields under different elevated $CO_2$ concentrations, Sci. Total Environ., 773, 145629, https://doi.org/10.1016/j.scitotenv.2021.145629, 2021.

Wu, K., Gong, P., Zhang, L., Wu, Z., Xie, X., Yang, H., Li, W., Song, Y., and Li, D.: Yield-scaled $N_2O$ and $CH_4$ emissions as affected by combined application of stabilized nitrogen fertilizer and pig manure in rice fields, Plant Soil Environ., 65, 497–502, https://doi.org/10.17221/286/2019-PSE, 2019.

Yang, S., Jiang, Z., Sun, X., Ding, J., and Xu, J.: Effects of Biochar Amendment on $CO_2$ Emissions from Paddy Fields under Water-Saving Irrigation, Int. J. Environ. Res. Public Health, https://doi.org/10.3390/ijerph15112580, 2018.

Yuan, M., Cai, C., Wang, X., Li, G., Wu, G., Wang, J., Geng, W., Liu, G., Zhu, J., and Sun, Y.: Warm air temperatures increase photosynthetic acclimation to elevated $CO_2$ concentrations in rice under field conditions, Field Crop. Res., 262, 108036, https://doi.org/10.1016/j.fcr.2020.108036, 2021.

Zhang, H., Yu, C., Kong, X., Hou, D., Gu, J., Liu, L., Wang, Z., and Yang, J.: Progressive integrative crop managements increase grain yield, nitrogen use efficiency and irrigation water productivity in rice, Field Crop. Res., 215, 1–11, https://doi.org/10.1016/j.fcr.2017.09.034, 2018.

Zhang, H., Liu, H., Hou, D., Zhou, Y., Liu, M., Wang, Z., Liu, L., Gu, J., and Yang, J.: The effect of integrative crop management on root growth and methane emission of paddy rice, Crop J., 7, 444–457, https://doi.org/10.1016/j.cj.2018.12.011, 2019.

Additionally, we have included the results of site validation in the revised manuscript as follows:

➢ The proposed rice calendar successfully extracts rice transplanting and harvest dates at 0.5° grid-cell across monsoon Asia by utilizing the rice feature-based phenology algorithm (Zhao et al., 2023) on Sentinel-1 and Sentinel-2 images (Fig. 2 Step 1 Algorithm a). The detected transplanting and harvest dates have been validated against 40 site-scale records from the literatures, showing high agreement with $R^2$ of 0.9 and 0.87, Bias of 7.99 and −9.07 days, MAE of 16.32 and 19.58 days, and RMSE of 19.00 and 22.43 days for transplanting and harvest days, respectively (Supplementary Text 3.1). The robustness of the site validation (Supplementary Text 3.1), combined with reasonable performance compared to other rice calendars (Fig. 9), further demonstrates the efficacy of the transplanting and harvest dates in the proposed rice calendar.

The main difference between the proposed rice calendar with other rice calendar lies in the algorithm for phenological date extraction. In contrast to census-based methods (such as the RiceAtlas rice calendar) that face the issue of overlapping rice croppings, and remote sensing-based methods (such as the RICA rice calendar) that rely on constant threshold values set for large areas, this algorithm is not limited by rice variety, management, and environmental factors. It extracts the features of flooding around the transplanting date and peak yellowness during harvest from the minimum VH and peak NDYI values, respectively, without setting threshold parameters to characterize rice phenological variation. Unfortunately, due to the absence of ground-truth data, it is not possible to validate

the Asian continental scale rice calendar with correct accuracy. Instead, the validation in this study was based on observational records available in the previous literature. In this validation, it is worth noting that the proposed rice calendar showed a relatively high coefficient of determination and low RMSE compared to other rice calendars (Supplementary Text 3.2) (Lines 397-414).

References:

Zhao, X., Nishina, K., Akitsu, T.K., Jiang, L., Masutomi, Y., Nasahara, K.N.: Feature-based algorithm for large-scale rice phenology detection based on satellite images, Agric. For. Meteorol., 329, 109283, https://doi.org/10.1016/j.agrformet.2022.109283, 2023

*R3C3: In Central China, there are could be bias in some regions with triple-cropping intensity. Please double check it.*

**Response:** Thank you for this valuable comments. We agree with you that the detection of triple cropping in central China could be bias, as most rice cultivation in central China follows a single or double rice cropping system (Liu et al., 2020; Zhang et al., 2020). The bias of triple cropping detection in central China might result from the presence of multiple crops cropping systems, leading to an overestimation of the number of rice croppings. In other words, before or after rice cultivation, other crops might be planted, which could be wrongly considered as one of the rice croppings. We have discussed this uncertainty in the "Uncertainty" section of the manuscript as follows:

"Furthermore, the complexity of multiple crop cropping systems can lead to an overestimation of the number of rice croppings. The growth of other crops exhibits a

similar pattern of a mono-peaked EVI time series and flood irrigation before sowing, similar to rice (Ahmad and Iram, 2023). Examples include the middle rice cropping system (rice with wheat, barley, or rapeseed cropping systems) in East and Central China (Chen et al., 2020) and the rice–wheat cropping systems on the Indo-Gangetic Plain (Abrol, 1997; Dhanda et al., 2022)."

To call attention to this bias, we have added the following text after the discussion regarding the uncertainty:

➢ Furthermore, the complexity of multiple crop cropping systems can lead to an overestimation of the number of rice croppings.  The growth of  other crop exhibits a similar pattern of a mono-peaked EVI time series and flood irrigation before sowing, similar to rice (Ahmad and Iram, 2023). Examples include the middle rice cropping system (rice with wheat, barley, or rapeseed cropping systems) in East and Central China (Chen et al., 2020) and rice–wheat cropping systems on the Indo-Gangetic Plain (Abrol, 1997; Dhanda et al., 2022). Thus, detected triple rice in central China (Fig. 10a) will be bias, which requires specific noted when using it. (Lines 474-479).

References:

Liu, L., Xiao, X., Qin, Y., Wang, J., Xu, X., Hu, Y., and Qiao, Z.: Mapping cropping intensity in China using time series Landsat and Sentinel-2 images and Google Earth Engine, Remote Sens. Environ., 239, 111624, https://doi.org/10.1016/j.rse.2019.111624, 2020.

Zhang, G., Xiao, X., Dong, J., Xin, F., Qin, Y., Doughty, R.B., and Moore, B.: Fingerprint of rice paddies in spatial–temporal dynamics of atmospheric methane concentration in monsoon Asia, Nat. Commun., 11, 554, https://doi.org/10.1038/s41467-019-14155-5, 2020.

*R3C4: According to Figure 1b, most of the 0.5º grids have a very low rice proportion, which are dominated by other crops than paddy rices, but the rice cropping intensity map shown in Figure 10 may not consider the potential rice and non-rice mixture issues.*

**Response:** We greatly appreciate the reviewer's helpful comments. The reviewer raises the issue that the rice cropping intensity map shown in Figure 10 may not consider the rice-crop mixing at low rice percentage grid. To be honest, it is not easy to depict the non-rice mixed with rice in the figure of rice cropping intensity. Inspired by your idea, we have recognized that the low rice proportion might have the risk of errors in identifying rice cropping intensity, that is, non-rice will be mistakenly identified as rice cultivation, or wrongly identified as one of rice croppings. Although we have tried to avoid this problem through sampling strategy – rice paddy fields were randomly selected from rice paddy field distribution map to obtain the rice phenology at each 0.5°grid. Obviously, the possibility of classification errors in the rice paddy field distribution map will increase at lower percentage rice grids, consequently resulting in the high possibility of non-rice crops being mixed with rice, as you mentioned. Therefore, to raise attention to this issue, we have incorporated this issue into the "Uncertainty" discussion section. Additionally, we have provided a potential solution for the future, involving the application of a relatively higher resolution rice distribution map. Please see the revision as follows:

➢ Furthermore, the complexity of multiple crop cropping systems can lead to an overestimation of the number of rice croppings.  The growth of  other crop exhibits a similar pattern of a mono-peaked EVI time series and flood irrigation before sowing, similar to rice (Ahmad and Iram, 2023). Examples include the middle rice cropping system (rice with wheat, barley, or rapeseed cropping systems) in East and Central China (Chen et al., 2020) and rice–wheat cropping systems on the Indo-Gangetic Plain (Abrol, 1997; Dhanda et al., 2022). Thus, detected triple rice in central China (Fig. 10a) will be bias, which requires

specific noted when using it. Except for the rice-predominant areas, the rice–crop mixing problem can also puzzle the grids with a low rice percentage. While rice phenology extraction was obtained through randomly selected sampling of rice paddy fields from the 500 m resolution rice distribution map (Zhang et al., 2020), grids with a low rice percentage have a higher possibility of errors in wrongly classifying non-rice crops as rice, consequently resulting in a higher possibility of non-rice crops being considered as rice cultivation or one of the rice croppings. The application of a higher resolution rice distribution map is expected to address this issue. (Lines 474-484).

*Minor comments:*

*R3C5: Line 77: "Normalized Yellow Index" should be "Normalized Differenced Yellow Index"*

**Response:** Our apologies. We have revised this as follows in the revised manuscript:

➢ A feature-based algorithm, proposed for large-scale rice phenology detection (Zhao et al., 2023), excels in utilizing backscattering (VH) and vegetation indices (Enhanced Vegetation Index (EVI) and Normalized Difference Yellow Index (NDYI) derived from Sentinel-1&-2 images to reflect features related to rice cultivation such as flooding, maximum leaf area, and most yellowness around transplanting, heading, and harvest date (Lines 75-78).

*R3C6: Figure 11 shows very significant differences among the four sources of rice cropping maps, more potential reason on the definition of different products should be carefully considered.*

**Response:** Thank you for your insightful comment. We agree with you that the more potential reasons among the rice calendars should be carefully considered to explain the observed significant differences in rice paddy field area, as shown in Fig. 11. One potential reason for these area differences could be the different methodologies used for rice calendar production. For instance, the RicaAtlas rice calendar is derived from sub-national statistics compilation, while the RICA and the proposed rice calendar are based on remote sensing techniques. Compared to the census-based method, remote sensing can capture the varying rice phenology at fine heterogeneity within administrative units. These methodological differences might lead to divergent representations of rice cropping areas. Additionally, the algorithm employed in retrieving phenological dates could affect the observed differences. The rule-based algorithm used in the RICA rice calendar relies on the turning point or key nodes of vegetation index, which are constant for large areas. The RiceAtlas rice calendar faces the problem of overlapping between croppings. Instead, feature-based algorithm used for the proposed rice calendar captures the flooding around transplanting date and peak yellowness during harvest. Furthermore, factors such as the spatial resolution and treatment of fragmented rice paddy fields could explain the area differences. For example, the spatial resolution of the RiceAtlas rice calendar is not uniform due to the sub-national statistics covering different spatial extents. All rice paddy fields in some sub-national areas were identified as single/double/triple croppings, and the area was calculated accordingly. Although the RICA rice calendar was produced by the satellite images, it was eventually converted to sub-national spatial scales. Given the coarse spatial resolution, fragmented rice paddy fields cannot be recognized in these three rice calendars either. Moreover, we conducted site validation among all the rice calendars. The site validation results reinforce the advantage of the proposed rice calendar. The phenological dates in the proposed rice calendar were found to be closer to the in-site records compared to those of other three rice calendars. The relatively large bias and variance in these three rice calendars further demonstrate their limitations and uncertainties in calculating the paddy area.

As mentioned in Section 3.3 'Advantages of the proposed rice calendar' and supported

by the newly added site validation results, we have revised the manuscript to address these reasons for the significant differences in paddy areas among the rice calendars as follows:

➢ The advantages of the above-mentioned algorithms (Fig. 2 Step 1, Step 2) largely contribute to the production of a gridded rice calendar. The proposed rice calendar provides spatially explicit rice phenology with continental coverage through remote sensing methods. The major difference between the proposed rice calendar and the RICA rice calendar lies in the use of a feature-based algorithm with VH and NDYI, which allows the proposed rice calendar to theoretically estimate rice phenology more accurately. Zhao et al. (2023) demonstrated that VH can accurately capture the start of paddy water logging, and NDYI is a good indicator of rice maturity stage. The proposed rice calendar presents a highly patchy map of rice phenological information (Figs.6 and 10a). The 0.5° resolution of the proposed rice calendar is finer than that of other rice calendars, including the RiceAtlas, RICA at sub-national scale, and SAGE derived from sub-national data. This improvement greatly reduces the bias error caused by assigning averaged rice phenology to administrative units, as rice phenology can vary considerably with large administrative units (Franch et al., 2022). Furthermore, the proposed rice calendar displays the detailed distribution of rice paddy fields (Figs. 6 and 10a), in contrast to previous rice calendars that covered entire administrative areas, irrespective of the small proportion of rice cultivation (Figs. S6-S8 and 10b-d). Site-scale validation reinforces the above-mentioned advantages, as the phenological dates in the proposed rice calendar are closer to the in-site records (Fig. S13; Fig. S14; Supplementary Text 3). The relatively large bias and variance in other three rice calendars (Fig. S14) demonstrate their limitations and uncertainties in calculating the rice paddy field area as shown in Fig. 11. (Lines 447-461).

*R3C7: Please check whether the data links are active, and hopefully the data and code links can be available later.*

**Response:** We appreciate the reviewer's suggestions regarding data accessibility. We have checked the data and code links (https://www.nies.go.jp/doi/10.17595/20230728.001-e.html) provided in the manuscript, and they are now active and accessible.